# Trimannose-coupled antimiR-21 for macrophage-targeted inhalation treatment of acute inflammatory lung damage

Christina Beck[1,2,12], Deepak Ramanujam [1,2,11,12], Paula Vaccarello [1], Florenc Widenmeyer [1], Martin Feuerherd[3], Cho-Chin Cheng[3], Anton Bomhard[1], Tatiana Abikeeva[1], Julia Schädler[4], Jan-Peter Sperhake[4], Matthias Graw[5], Seyer Safi[6], Hans Hoffmann[6], Claudia A. Staab-Weijnitz [7], Roland Rad [8], Ulrike Protzer [3,9], Thomas Frischmuth[10,11] & Stefan Engelhardt [1,2] ✉

Recent studies of severe acute inflammatory lung disease including COVID-19 identify macrophages to drive pulmonary hyperinflammation and long-term damage such as fibrosis. Here, we report on the development of a first-in-class, carbohydrate-coupled inhibitor of microRNA-21 (RCS-21), as a therapeutic means against pulmonary hyperinflammation and fibrosis. MicroRNA-21 is among the strongest upregulated microRNAs in human COVID-19 and in mice with acute inflammatory lung damage, and it is the strongest expressed microRNA in pulmonary macrophages. Chemical linkage of a microRNA-21 inhibitor to trimannose achieves rapid and specific delivery to macrophages upon inhalation in mice. RCS-21 reverses pathological activation of macrophages and prevents pulmonary dysfunction and fibrosis after acute lung damage in mice. In human lung tissue infected with SARS-CoV-2 ex vivo, RCS-21 effectively prevents the exaggerated inflammatory response. Our data imply trimannose-coupling for effective and selective delivery of inhaled oligonucleotides to pulmonary macrophages and report on a first mannose-coupled candidate therapeutic for COVID-19.

Destruction of pulmonary cells by infection or in response to pulmonary-toxic substances triggers a massive immune response that by itself promotes further lung damage and may progress to a condition termed Acute Respiratory Distress Syndrome (ARDS). Despite some progress in the management of ARDS, its lethality remains high with treatment options being limited[1,2]. During the recent surge in studies on lung injury and ARDS in COVID-19, the pulmonary macrophage has been identified as the key cell type that mediates the pathologically exaggerated immune response, also termed hyperinflammation[3–6]. It is the extent and duration of this exaggerated immune response that largely determines the clinical course of ARDS caused by acute inflammatory lung damage[7]. Damage-associated

[1]Institute of Pharmacology and Toxicology, Technical University of Munich (TUM), Munich, Germany. [2]DZHK (German Centre for Cardiovascular Research), partner site Munich Heart Alliance, Munich, Germany. [3]Institute of Virology, Helmholtz Munich, Technical University of Munich (TUM), School of Medicine, Munich, Germany. [4]Institute of Legal Medicine, University Medical Center Hamburg-Eppendorf, Hamburg, Germany. [5]Institute of Legal Medicine, Faculty of Medicine, Ludwig-Maximilians-Universität (LMU) München, Munich, Germany. [6]Division of Thoracic Surgery, Klinikum rechts der Isar, Technical University of Munich (TUM), Munich, Germany. [7]Comprehensive Pneumology Center, Institute of Lung Health and Immunity, Helmholtz Center Munich, Member of the German Center of Lung Research (DZL), Munich, Germany. [8]Institute of Molecular Oncology and Functional Genomics, Translatum Cancer Center, School of Medicine, Technical University of Munich (TUM), Munich, Germany. [9]German Center for Infection Research (DZIF), partner site Munich, Munich, Germany. [10]Baseclick GmbH, Neuried, Germany. [11]RNATICS GmbH, Planegg-Martinsried, Germany. [12]These authors contributed equally: Christina Beck, Deepak Ramanujam. ✉e-mail: Stefan.engelhardt@tum.de

molecular patterns (DAMPs) released from dying cells appear to constitute the principal activating signals that bind to a variety of receptors that mediate macrophage overactivation[5]. These DAMPs include a wide array of molecules ranging from small molecules to nucleic acids, proteins and debris from cells or organelles and activate a diverse set of receptors and receptor-like proteins on macrophages[8]. Further complicating effective therapeutic intervention, recent data indicate that, once initiated, pulmonary hyperinflammation in COVID-19 is self-sustained and driven by feed forward loops[9]. It is against this background, that strategies are urgently needed that interfere with pathological overactivation of pulmonary macrophages and thus with the pathophysiology leading to ARDS.

MicroRNAs are a class of small regulatory RNAs that have been shown to regulate multiple cellular functions and have recently emerged as promising drug targets[10]. We explored the miRnome of cardiac macrophages and found microRNA-21 (miR-21) to be the single strongest expressed microRNA in these cells[11]. Furthermore, we found macrophage miR-21 to mediate the pathology of inflammation-related fibrotic tissue remodelling in a cardiac disease model[11]. In the present study, we sought to map the entire miRnomes of murine and human lung in health and acute inflammatory lung disease. MiR-21 was the strongest expressed microRNA in pulmonary macrophages and one of the most upregulated miRNAs in lung tissue from a mouse model of acute inflammatory lung damage and from humans with COVID-19. In an effort to effectively interfere with miR-21 specifically in pulmonary macrophages, we developed a carbohydrate-coupled miR inhibitor of miR-21 (RCS-21). RCS-21 achieved rapid and strong uptake into pulmonary macrophages upon inhalation. We further report on its therapeutic efficacy in a mouse model of acute inflammatory lung damage and its ability to suppress the inflammatory response in human lung tissue infected with SARS-CoV-2.

## Results

### MiR-21 is highly expressed in pulmonary macrophages and upregulated in acute inflammatory lung disease, including COVID-19

We performed next-generation sequencing in human COVID-19 lungs (Supplementary Table 1) and in lungs from mice subjected to bleomycin-induced lung injury to determine their microRNA and transcriptome signatures (Fig. 1a). Among the top expressed microRNAs, we identified miR-21-5p as the most upregulated microRNA in both mouse lung after acute injury and human COVID-19 (Fig. 1b, c and Fig. S1a). Quantitative analysis of the distribution of mRNA fold changes of all genes revealed a significant repression of miR-21 binding site-carrying mRNAs (miR-21 targets), indicating increased activity of miR-21 in COVID-19 (Fig. S1b). Small RNA sequencing of different pulmonary cell types from human and mouse revealed miR-21-5p as the single highest expressed and enriched microRNA in macrophages (Fig. 1d and Fig. S2a–e). Immunostaining of lung sections furthermore revealed a strong increase of macrophage numbers in mice after acute lung injury and in humans with COVID-19 (Fig. 1e and Fig. S1a, confirming previous studies[3,4]). Within the increased macrophage population, we further observed a relative increase of recruited alveolar and interstitial macrophages upon bleomycin-induced injury in mice and SARS-CoV-2 infection in humans (Fig. S3a–f). In agreement with the small RNA sequencing data, in situ hybridization found miR-21 to be enriched in human lung macrophages and to be upregulated in this cell type in COVID-19 (Fig. S1a).

To examine miR-21 as a potential therapeutic target in pulmonary macrophages in vivo, we generated macrophage-specific miR-21–deficient mice (miR-21 cKO) by crossing miR-21–floxed mice with a mouse line that expresses Cre recombinase under the control of the Cx3cr1 promotor[12]. MiR-21 cKO mice exhibited an effective knockout of miR-21, as evidenced by a > 90% reduction of miR-21 levels in pulmonary macrophages (Fig. S4a).

To investigate whether macrophage miR-21 determines the development of inflammatory lung disease, we subjected miR-21 cKO mice to bleomycin-induced lung injury. Lung function was assessed at 14 days after bleomycin injury (Fig. 1f). WT mice treated with bleomycin showed typical impairment of pulmonary function (Fig. 1g, h and Fig. S4b). Bleomycin-treated miR-21 cKO mice displayed significantly better pulmonary function than wildtype littermate controls (Fig. 1g, h and Fig. S4b). Staining of extracellular matrix proteins in lung cross-sections indicated that in the miR-21 cKO group fibrosis had been significantly prevented, compared to bleomycin-treated WT littermate mice (Fig. 1i). Together, these findings prompted us to pursue macrophage miR-21 as a potential therapeutic target in acute inflammatory lung damage including that observed in COVID-19.

### Development of macrophage-targeted carbohydrate-coupled antimiR-21 (RCS-21)

We next aimed to identify candidate cell surface molecules on pulmonary macrophages that may allow cell type-specific targeting and delivery of microRNA inhibitors to these cells (see Fig. 2a for candidate stratification). Transcriptome analyses of the major pulmonary cell types in mouse lung identified mannose receptor 1 (*Mrc1*), macrophage receptor with collagenous structure (*Marco*) and Axl receptor tyrosine kinase (*Axl*) as the cell surface molecules with the highest combined ranks for macrophage abundance and specificity (Fig. 2b). Out of these we identified *Mrc1* to be the strongest expressed in both interstitial and alveolar macrophages (Fig. 2c and see Fig. S5a for comparison to other pulmonary cells). MRC1 is a transmembrane glycoprotein that allows macrophages to bind to carbohydrate moieties present on the surface of various bacteria and fungi and phagocytose the latter[13]. Analysis of single-cell transcriptomes of mouse lungs subjected to a model of acute inflammatory lung damage (bleomycin instillation) and of human lungs from subjects with COVID-19 demonstrated robust and selective expression of *Mrc1* in macrophages, which was maintained in disease (Fig. 2d, e) and independent from sex (Fig. S5b). In addition, immunofluorescent staining on lung cryosections confirmed an increase of cells staining positive for MRC1 in mice treated with bleomycin compared to controls (Fig. 2f). *MRC1* also showed exceptionally high expression in macrophages compared to other mannose-binding lectins (Fig. S6). We next aimed to exploit the selective expression of MRC1 and the high capacity uptake mediated by this receptor[13] as a means to deliver ligand-coupled oligonucleotide inhibitors of miR-21 selectively to macrophages (see Fig. 2h for uptake mechanism). To this end, we generated a locked nucleic acid (LNA)/deoxyribonucleic acid mixmer of antimiR-21 coupled with a chemical PEG-linker and trimannose (hereby referred to as RCS-21, Fig. 2g and Fig. S7). We selected trimannose as a carbohydrate ligand based on protein docking analyses (ProteinsPlus analysis platform[14]) to predict binding of different mannoses and mannans to MRC1. These analyses yielded higher binding affinity to MRC1 for trimannose in comparison to mono- and dimannose (Fig. S8), which is in line with experimental data[15]. Among the trimannoses, the branched form showed higher affinity compared to the linear form (Fig. S8). As a comparator ligand, we chose another multivalent sugar (here: triantennary N-acetylgalactosamine (GalNAc)) that has become a well-established delivery principle of oligonucleotides to the liver[16].

### Inhaled RCS-21 is specifically delivered to pulmonary macrophages

Unconjugated antimiR-21, RCS-21 and GalNAc-21 were tested in vivo for their quantitative uptake in pulmonary cell types by coupling to fluorescein-amidite dye (FAM) and the cellular uptake was analysed by flow cytometry. Indicated doses (0.625, 1.25 and 2.5 mg/kg) of oligonucleotides were administered into wild-type mouse lungs by aerosolized inhalation using a dedicated mouse nebulizer (flexiVent system; Fig. 3a). Unconjugated LNA-antimiR-21 served as control while

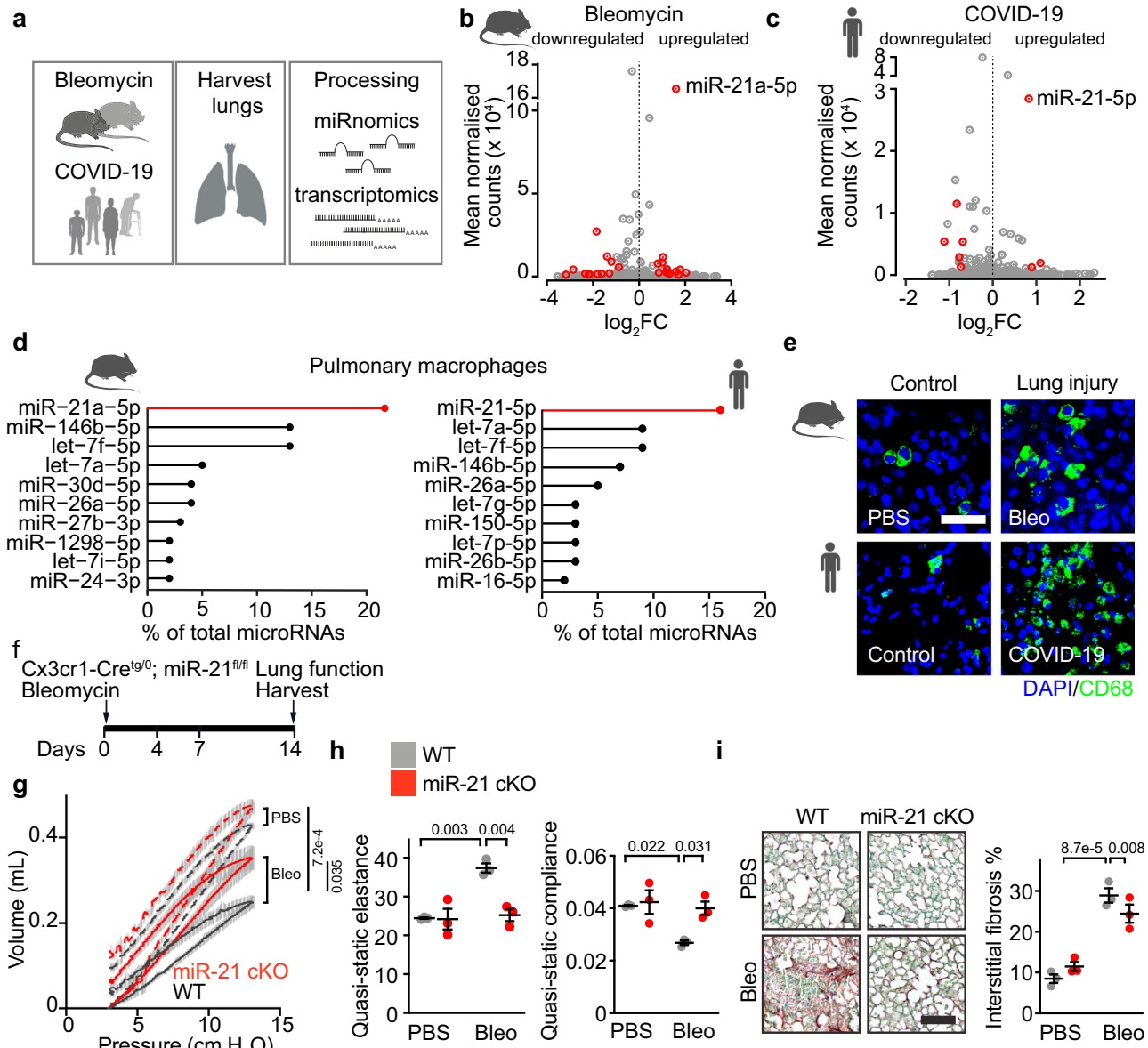

**Fig. 1 | Identification of miR-21 in macrophages as therapeutic target for acute inflammatory lung disease. a** Overview of the study design. **b** Scatter plot of differentially expressed microRNAs in bleomycin lungs compared to control lungs. Each dot represents one microRNA. Differentially expressed microRNAs are highlighted in red (FDR < 0.05, −0.75 < $\log_2$FC > 0.75, mean normalised counts > 1000). Control $n = 3$, Bleomycin $n = 3$. **c** Scatter plot of differentially expressed microRNAs in COVID-19 lungs compared to control lungs. Each dot represents one microRNA. Differentially expressed microRNAs are highlighted in red (FDR < 0.05, −0.75 < $\log_2$FC > 0.75, mean normalised counts > 1000); Control $n = 12$, COVID-19 $n = 14$. **d** Lollipop graph represents the top 10 expressed microRNAs in (left) mouse and (right) human lung macrophages ($n = 3$-4). **e** Representative staining for macrophage marker CD68 in bleomycin-induced mouse lungs (n(PBS) = 3, n(bleo) = 3) and COVID-19 patients (n(COVID-19) = 3, n(Control) = 3). Scale bar represents

50 μm. **f** Experimental strategy. Wild-type and macrophage-specific miR-21 deficient (miR-21 cKO) mice were administered with PBS or 2 U/kg bleomycin (bleo) into the lungs using a micro sprayer. Two weeks later, lung function was assessed, and lungs were harvested for morphometry and isolation of cells. **g** Mean tracings of pressure-volume curves. Gray shaded area behind the curves indicates standard error mean. PBS and bleomycin treatment are represented by dashed and solid lines, respectively. **h** Lung function as indicated by quasi-static elastance and quasi-static compliance. **i** Sirius red/fast green staining of representative lung sections and quantification. Scale bar represents 100 μm. **f–i** WT PBS $n = 3$, miR-21 cKO PBS $n = 3$, WT bleo $n = 3$, miR-21 cKO bleo $n = 3$. Data are mean ± SEM and individual values and were analysed using 2-way ANOVA with Tukey's post-test (two-sided). Source data are provided as a Source Data file.

PBS served as negative control. Quantitative assessment of RCS-21-FAM indicated superior delivery of these molecules to pulmonary macrophages compared to both GalNAc-coupled antimiR-21 and unconjugated antimiR-21 (Fig. 3b). Moreover, the absolute intensity of FAM signals in RCS-21-FAM-inhaled mice achieved higher fluorescence intensity per cell, indicative of higher intracellular concentrations of RCS-21 in pulmonary macrophages (Fig. 3c). Immunodetection of FAM signals in lung cryosections corroborated the highly specific delivery of RCS-21 to pulmonary macrophages (Fig. 3d). Analysis of inhaled RCS-21-FAM in other organs indicated no detectable signals in the heart, kidney and spleen but faint signals in interstitial cells of the liver which co-stained for the macrophage marker CD68 (Fig. S9). Taken together, aerosolized inhalation of RCS-21 resulted in selective and efficient delivery to pulmonary macrophages.

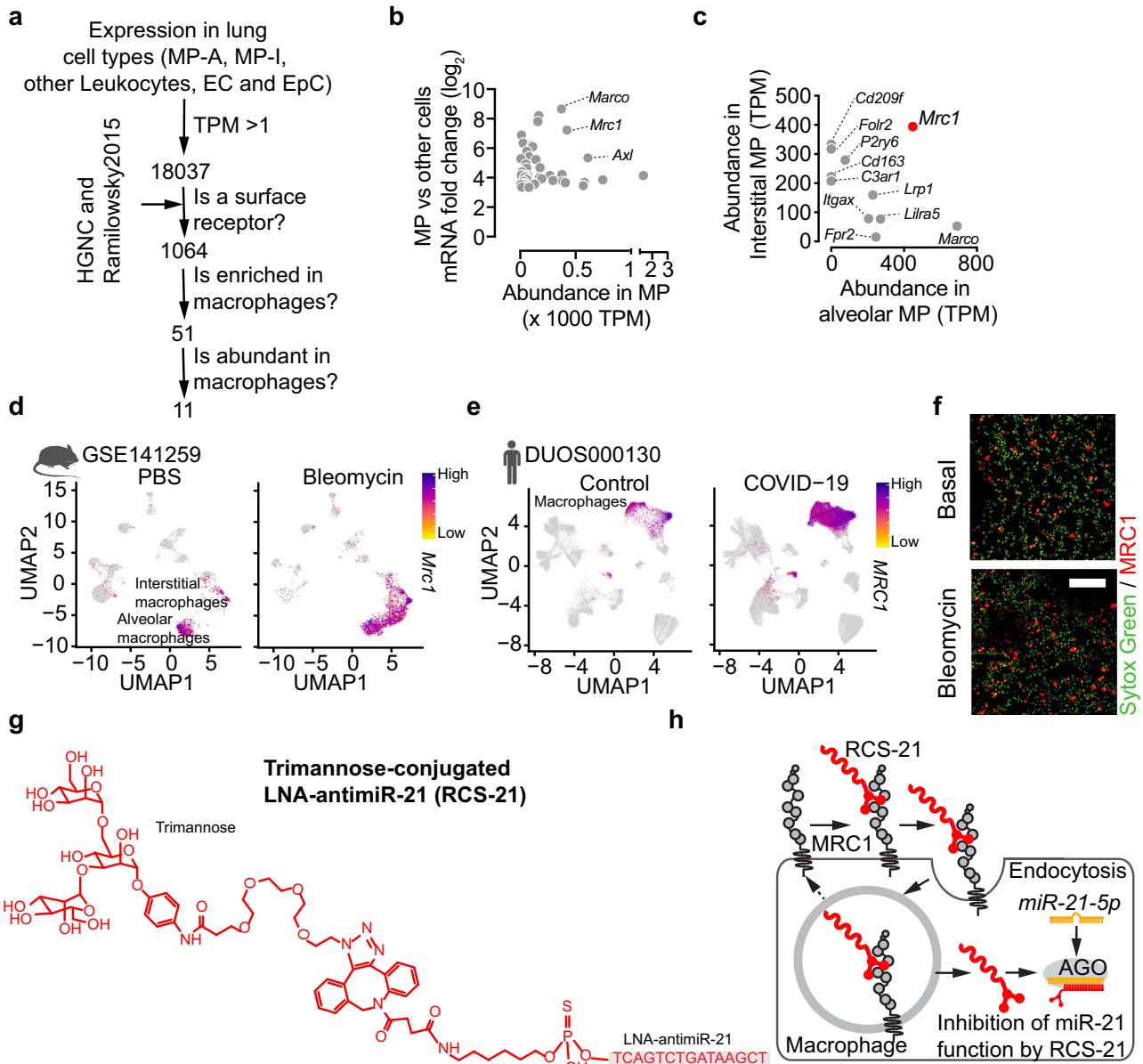

**Fig. 2 | Development of macrophage-targeted, carbohydrate-coupled antimiR-21 (RCS-21). a** Screening strategy to identify surface receptor genes that are enriched in macrophage populations compared to other cell populations in the lung (mRNA log$_2$ fold change > 3 and FDR < 0.05) and are abundant in both interstitial and alveolar macrophages (TPM > 100 in MP-A and MP-I and TPM < 10 in other cells). $n = 3$ per cell type. **b** Scatter plot showing surface receptor genes that are enriched in macrophage populations compared to other cell populations in the lung. **c** Comparative analysis identified mannose receptor 1 (MRC1) as the surface receptor enriched and abundantly expressed in both alveolar and interstitial lung macrophages. **d** Feature plot of *Mrc1* in mouse lungs after bleomycin injury (data from GSE141259). **e** Feature plot of *MRC1* in human lungs after COVID-19 (data from DUOS-000130). **f** Representative immunofluorescent staining for MRC1 in PBS- and bleomycin-treated mouse lungs (staining was carried out for three mice of each group). Nuclei were stained with Sytox Green. Scale bar represents 100 μm. **g** Chemical structure of trimannose-conjugated LNA-antimiR-21 (RCS-21). **h** Schematic illustration of uptake of RCS-21 by MRC1 in macrophages. MP-A alveolar macrophages, MP-I interstitial macrophages, EC endothelial cells, EpC epithelial cells. Source data are provided as a Source Data file.

## Inhibition of miR-21 reduced bleomycin-induced pulmonary remodelling and dysfunction

To test the therapeutic efficacy of RCS-21 in vivo, we subjected wild-type mice to acute lung injury by bleomycin, and a single inhaled dose of 50 μg (2.5 mg/kg body weight) of RCS-21 was administered using a nebulizer at 4 days after injury (assuming a deposition fraction in the lungs at 25%, 200 μg of RCS-21 was loaded onto the nebulizer) (Fig. 4a and see Fig. S10a for antimiR sequence). In alveolar cells obtained by lavage (i.e. BALF) at day 14 we found a highly significant reduction of miR-21 (−58%) in RCS-21 treated mice compared to control mice (Fig. S10b). In control treated animals, lung function assessed at 14 days after injury was markedly impaired with reduced pressure-volume relationship, decreased inspiration capacity, increased elastance and decreased compliance (Fig. 4b, c and Fig. S10c). Inhaled RCS-21 improved adverse pulmonary remodelling (Fig. 4b, c and Fig. S10c). There was no drug-related mortality or morbidity observed in the bleomycin or PBS groups treated with RCS-21. To correlate morphological and molecular differences with pulmonary function, the total mouse lung tissue was dissected into several parts for use in different assays (see scheme in Fig. S11a). In line with the functional data, pronounced adverse lung remodelling was observed in control-treated mice, evident as a significant increase in lung fibrosis and inflammation (Fig. 4d, e and Fig. S10d). Both lung fibrosis and inflammation were significantly reduced in RCS-21-treated mice (Fig. 4d, e and Fig. S10d).

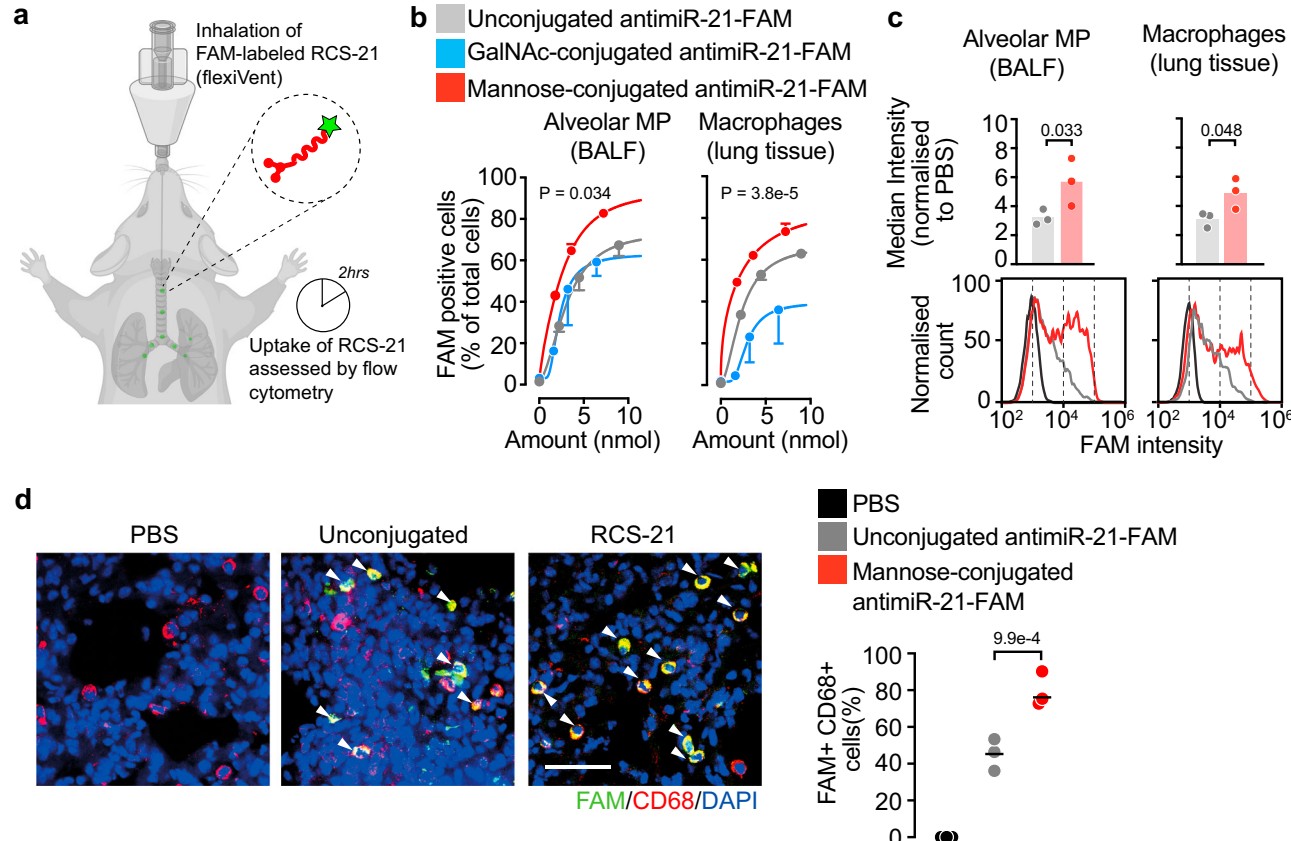

**Fig. 3 | Delivery of inhaled RCS-21 to pulmonary macrophages in vivo. a** Wild-type mice were administered with either unconjugated, N-acetylgalactosamine-(GalNAc) or trimannose-conjugated (RCS-21) antimiR-21-FAM by inhalation. Two hours later, mice were sacrificed and cells isolated from bronchoalveolar lavage fluid (BALF) and lung tissue were assessed for FAM signals by flow cytometry. **b** Percentage of FAM-positive cells in different lung macrophage fractions. Unconjugated $n = 3$ per group, RCS-21 $n = 3$ per group, GalNAc-conjugated $n = 2$ per group. Data are mean ± SEM and asymmetric nonlinear regression analysis was used for curve fitting and compare the groups. **c** Top, median fluorescence intensity of FAM signals in macrophages. Bottom, representative histogram of

macrophage subsets. 1.25 mg/kg Unconjugated $n = 3$ and 1.25 mg/kg RCS-21 $n = 3$. Data are mean and individual values, and were analysed using two-sided Student's $t$-test. **d** Left, representative immunofluorescent staining of 5 µm mouse lung tissue cryosections for CD68 as a marker for macrophages. Nuclei were stained with DAPI. Scale bar represents 50 µm. White arrow indicates FAM-positive macrophages. Right, quantification of the same. PBS $n = 3$, 1.25 mg/kg Unconjugated $n = 3$ and 1.25 mg/kg RCS-21 $n = 3$. Data are mean and individual values and were analysed using two-sided one-way ANOVA with Sidak's post-test. FAM fluorescein amidites, MP macrophages. Source data are provided as a Source Data file.

Single cell sequencing was then carried out on primary lung cells isolated from RCS-21 and control-treated mice 14 days after bleomycin. Cells were multiplexed from three different animals in each group and the proportion of CD45+ cells was reduced to 50% to avoid over-sampling (Fig. S11a, b). MACS-based separation of CD45+ leucocytes corroborated their increase in bleomycin-induced lung injury (Figs. S11c).

Transcriptional profiles of > 7500 single cells per group were obtained using the 10X Chromium platform and yielded 20 different clusters subdivided into 7 major cell populations (Figs. S11d–h). Based on established markers[17] we further subclustered the macrophage population and identified resident alveolar macrophages ($Marco^+$ / $Mrc1^{hi}$ / $Ccr2^-$ / $Itgam^{lo}$ / $Itgax^{hi}$), recruited alveolar macrophages ($Marco^-$ / $Mrc1^{lo}$ / $Ccr2^{hi}$ / $Itgam^-$ / $Itgax^-$) and interstitial macrophages ($Marco^-$ / $Mrc1^{lo}$ / $Ccr2^{hi}$ / $Itgam^{hi}$ / $Itgax^-$) (Fig. S12a, b). RCS-21 treatment led to a relative decrease of monocytic-derived recruited alveolar macrophages and an increase in alveolar macrophages (Fig. S12c). In addition, analysing the absolute number of significant differentially expressed genes, the alveolar macrophage population showed the strongest transcriptome deregulation among others, illustrating the importance of this cell type in disease pathogenesis as well as in RCS-21 treatment (Fig. S12d). To assess miR-21 activity, we conducted comparative transcriptome analysis of mRNAs harbouring predicted

binding sites for miR-21 (termed miR-21 targets) and mRNAs devoid of miR-21 bindings sites (non-targets). This revealed selective and significant de-repression of miR-21 targets in the alveolar and interstitial macrophage populations in response to treatment with RCS-21 (Fig. S12e). Gene ontology enrichment analysis of the top 200 macrophage mRNAs de-regulated after lung injury showed enrichment of processes related to inflammatory and innate immune responses, which was reversed after inhalation of RCS-21 (Figs. S13a, b). Likewise, after RCS-21 treatment alveolar macrophages showed a pronounced reversal of the bleomycin injury-associated changes (Fig. S13c). In agreement with the observed anti-fibrotic effect of RCS-21, the fibroblast clusters showed a reduction of extracellular matrix genes including $Col1a1$, $Col1a2$ and $Col3a1$ (Fig. S14).

## Therapeutic efficacy of RCS-21 in SARS-CoV-2-infected human lung tissue

To study the therapeutic effect of RCS-21 in a human model of COVID-19, human precision-cut lung slices (hPCLS) (Fig. S15) were infected with SARS-CoV-2 and 96 h later tissues were harvested (Fig. 5a and Supplementary Table 2). In preparatory experiments, we first assessed the uptake of RCS-21 in native hPCLS, employing fluorescently (FAM)-labelled RCS-21 and found preferential uptake of RCS-21 to MRC1+ cells (Figs. S16a, b). Successful infection was

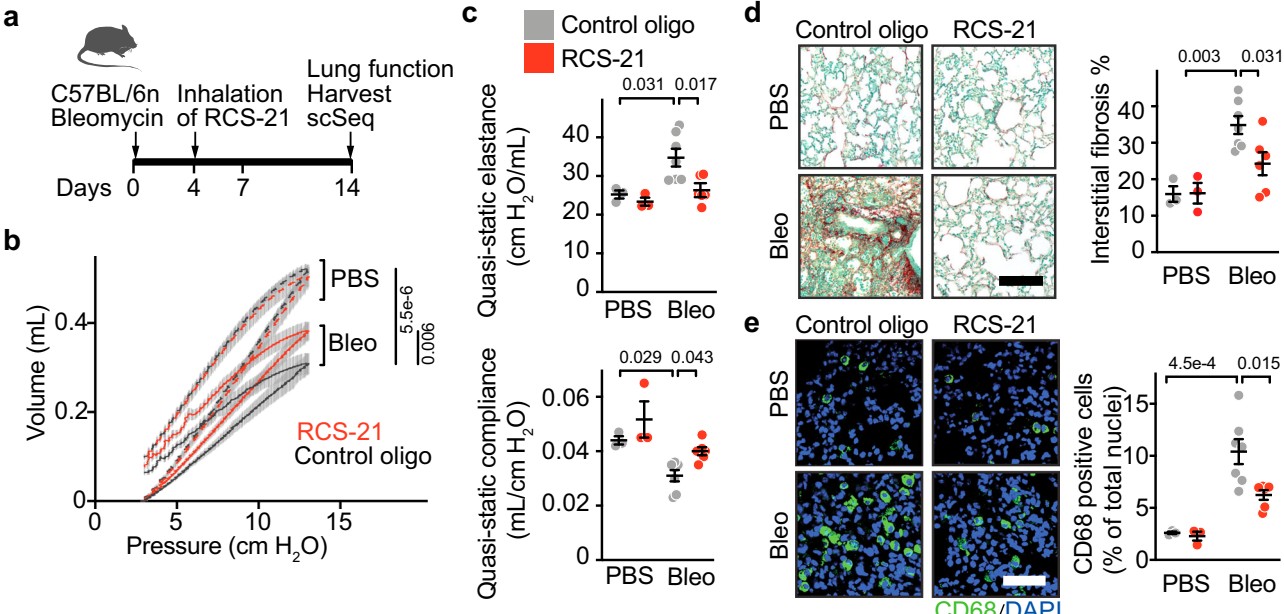

**Fig. 4 | Therapeutic efficacy of inhaled RCS-21 in a preclinical animal model of acute inflammatory lung damage. a** Experimental strategy. Wild-type mice were administered with PBS or 2 U/kg bleomycin (bleo) into the lungs using a micro sprayer. 2.5 mg/kg RCS-21 or control oligo (containing mismatch-miR-21 sequence) was applied by inhalation 4 days after injury. 10 days later, lung function was assessed, and lungs were harvested for morphometry and isolation of cells. **b** Mean tracings of pressure-volume curves. Gray shaded area behind the curves indicates standard error mean. PBS and bleomycin treatment is represented by dashed and solid lines, respectively. **c** Lung function as indicated by quasi-static elastance and quasi-static compliance. **d** Left, representative staining and analysis of lung sections using Sirius red/Fast green. Scale bar represents 100 μm. Right, quantification of fibrosis. **e** Left, representative staining of CD68. Right, quantification of CD68 positive cells. Scale bar represents 50 μm. **b**–**e** Control oligo: PBS *n* = 3, bleo *n* = 7. RCS-21: PBS *n* = 3, bleo *n* = 6. Data are mean ± SEM and individual values and were analysed using two-way ANOVA with Tukey's post test (two-sided). Source data are provided as a Source Data file.

next validated by aligning the transcriptome reads to the SARS-CoV-2 reference genome (Fig. 5b). Upon infection with SARS-CoV-2, hPCLS exhibited upregulation of miR-21-5p comparable to what we had observed in human lung tissue from individuals with COVID-19 (Fig. 5c, d). Treatment with RCS-21 24 h after SARS-CoV-2 infection inhibited miR-21 as detected by immunofluorescence (Fig. 5c) and small RNASeq (Fig. 5d) almost completely. Then, deep RNA sequencing of the same samples of hPCLS was carried out to determine the pathological transcriptome signature induced by SARS-CoV-2 and to test for a potential therapeutic effect of RCS-21 (Fig. 5e–i). Principal component analysis revealed distinct clustering of infected hPCLS that was clearly separated from uninfected samples and that was partially normalised by RCS-21 (Fig. 5e). Similar to what we had initially observed in the post-mortem COVID-19 lung samples (Fig. S1b), also the hPCLS system showed a significant repression of predicted miR-21 targets upon infection with SARS-CoV-2, indicating increased activity of miR-21 (Fig. S17). Treatment with RCS-21 effectively interfered with miR-21 activity as evidenced by a significant de-repression of miR-21 targets (Figs. 5f, g). Infection of hPCLS by SARS-CoV-2 induced a prominent pro-inflammatory gene signature (containing many known SARS-CoV-2-associated chemokines and pro-inflammatory cytokines), which was almost completely reversed by treatment with RCS-21 (Fig. 5h). Treatment with RCS-21 had no significant effect on SARS-CoV-2 reads (Fig. S18), suggesting a mechanism independent of the viral load but rather through a downstream mechanism such as the anti-inflammatory effect we observed in mice. Gene ontology analysis of whole transcriptomes confirmed that the gene expression of infected hPCLS was dominated almost completely by inflammation-related pathways (Fig. 5i). Again, treatment with RCS-21 drastically reduced this pro-inflammatory signature in this human model of COVID-19 (Fig. 5i).

## Discussion

In the present study we report on the development of a first-in-class, carbohydrate-coupled microRNA oligonucleotide drug that interferes with pathological macrophage activation in acute inflammatory lung disease. RCS-21 is directed against miR-21, the strongest expressed microRNA in macrophages, where it is required for pro-inflammatory polarization as well as pro-fibrotic activation of adjacent fibroblasts[11]. RCS-21 exerts a number of remarkable characteristics, three of which we briefly discuss in the following.

(1) RCS-21 possesses a high degree of target cell specificity, achieved by i) the high cell type specificity of the cognate receptor MRC1 on alveolar macrophages, a drug targeting receptor well characterized for small compound and recently siRNA therapeutics[13,15,18], ii) the application via inhalation that reaches the target cells directly and in high concentration while minimizing systemic exposure.

(2) Independence of COVID-variants. The rapid and unforeseen evolution of new variants of SARS-CoV-2[19] poses a particular problem to the management of the current pandemic and challenges both vaccination and anti-viral strategies[20]. Such variants of concern may be characterized by particularly severe disease and/or high transmissibility and they must be expected to lead to future high prevalence of severe COVID-19 and pulmonary affection[20]. Due to its mechanism of action that targets the common, downstream hyperactivation of pulmonary macrophages, we are tempted to speculate that the therapeutic efficacy of RCS-21 may prove independent of current and future variants of SARS-CoV-2. Of note, our data on the Omicron variant in cultivated human lung tissue infected with this variant support this view.

(3) Anti-fibrotic efficacy. Our data in mice suggest that RCS-21 is also effective beyond the immediate re-polarization of hyperactivated pulmonary macrophages by preventing pulmonary dysfunction

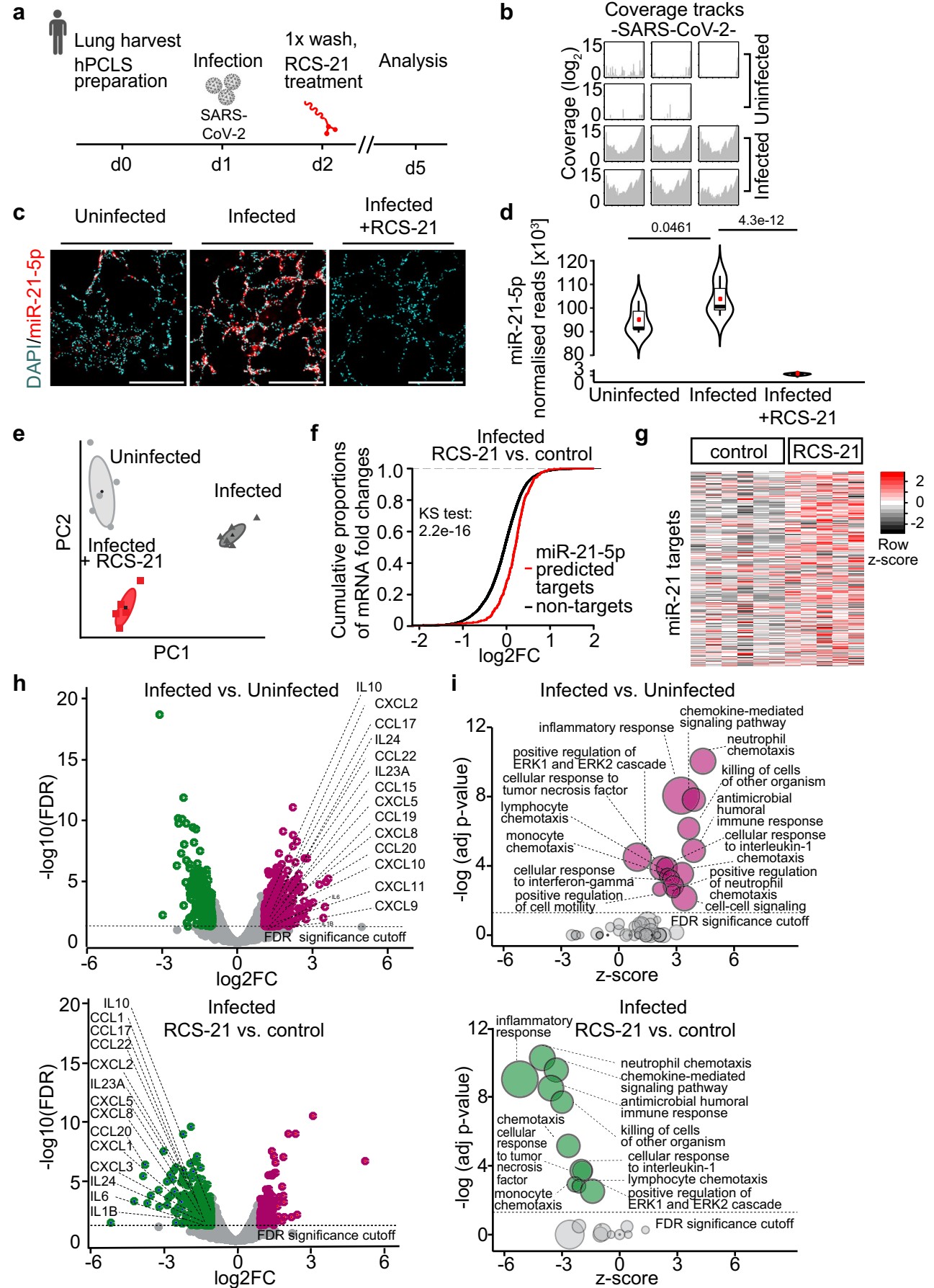

**Fig. 5 | RCS-21 inhibits inflammatory activation of human lung tissue in response to SARS-CoV-2 (Omicron). a** Overview of the study. **b** RNA-Seq of uninfected and infected hPCLS. BAM files were aligned onto the SARS-CoV-2 reference genome. **c** MicroRNA Red Scope staining for hsa-miR-21-5p in hPCLS (blue: nucleus). Scale bar represents 200 μm. Representative of $n = 2$ individuals per condition. **d** DESeq2 normalised reads of miR-21-5p in indicated hPCLS groups. Box plots represent the 25th and the 75th percentiles. The horizontal line represents the median, the red dots represent the mean for each group. Statistical analysis was performed using two-sided one-way ANOVA with Tukey's post test. **e** Principal component analysis of the transcriptomes. **f** Cumulative distribution curves of mRNA transcriptomes to asssess miR-21-5p activity. Rightward shift of predicted miR-21 targets (red) indicates decreased miR-21 activity (RCS-21 vs. control). Statistical analysis was performed using two-sided Kolmogorov-Smirnov (KS) test. **g** Heatmap of conserved predicted targets of miR-21. **h** Volcano plots representing differentially expressed genes between infected and uninfected hPCLS (upper panel) and between infected RCS-21-treated and infected control hPCLS (lower panel). Significantly upregulated genes ($log_2FC > 1$, FDR < 0.05) are highlighted in violet and significantly downregulated genes ($log_2FC < -1$, FDR < 0.05) in green. **i** GO bubble plots represent the biological processes for upregulated ($log_2FC > 1$) and downregulated genes ($log_2FC < -1$) between SARS-CoV-2 infected and uninfected hPCLS (upper panel) and between infected RCS-21-treated and infected control hPCLS (lower panel), FDR significance cutoff < 0.01, upregulated pathways are highlighted in violet, downregulated pathways are highlighted in green. As a statistical test DAVID uses a modified Fisher's Exact test; **d**–**i** $n$(uninfected) = 5, $n$(infected) = 6, $n$(RCS-21) = 5. Source data are provided as a Source Data file.

and fibrosis. The latter is one of the hallmarks of pulmonary disease in long-COVID and there is currently no effective medication with regard to this much-feared long-term sequelae in survivors of severe COVID-19[21]. We had recently described that it is miR-21 in myocardial macrophages that determines fibroblast activation and fibrosis in the mammalian heart[11]. Remarkably, recent studies on COVID-19 likewise identified the pulmonary macrophage as responsible for pulmonary fibrosis[6]. These findings and our data on human lung slices infected with SARS-CoV-2 strongly suggest that we may expect therapeutic efficacy of RCS-21 against pulmonary fibrosis in patients with COVID-19.

## Therapeutic perspectives for trimannose-coupled nucleic acid therapeutics

Our development of trimannose-coupled antimiR-21 to target pulmonary macrophages resembles the effective targeting of liver cells using coupling of siRNAs to the carbohydrate GalNAc, that is meanwhile used in several approved drugs[16]. Similar to the GalNAc principle for targeting hepatocytes, we propose that trimannose-coupling of nucleic acids should be generally applicable for targeting macrophages beyond miR-21 and in other organs.

## Methods

### Post mortem sampling and ethics statement

Autopsies from non-COVID-19 patients and COVID-19 patients were performed at the Institute of Legal medicine at the Ludwig-Maximilians-Universität München (LMU, Munich, Germany) or at the Institute of Legal medicine at the University Medical Center Hamburg-Eppendorf (UKE, Hamburg, Germany). Lung sections from the lower left lung lobe were taken maximum 48 h post mortem. For all the experiments an ethics committee approval (59/21 S, Ethikkommission der Fakultät für Medizin) from the Technical University of Munich (TUM) and a signed written informed consent from the closest relatives were obtained.

### Human donors and ethics statement for preparing precision-cut lung slices

Tumour-free lung sections from cancer patients who underwent surgery at the Klinikum rechts der Isar (Munich, Germany) were received. For all the experiments an ethics committee approval (59/21 S) from the Technical University of Munich and a signed written informed consent from the patients were obtained.

### Experimental animals and study design

Cx3cr1-Cre[Tg/0] and miR-21-floxed mice have been described previously[12]. In brief, macrophage-specific miR-21–deficient mice (miR-21 cKO) were generated by crossing miR-21–floxed mice with *Cx3cr1*-Cre tg/0 mice (Stock: 025524, The Jackson Laboratory), a mouse line that expresses Cre recombinase under control of the *Cx3cr1* promotor that is specifically active in macrophages. All mice were bred in C57BL/6N genetic background. All animal studies were performed in accordance with relevant guidelines and regulations of the responsible authorities and approval was obtained from the IRB at the Regierung von Oberbayern (ROB-55.2-2532.Vet_02-19-82). Mice were housed in rooms maintained at constant temperature (25 °C) and humidity (45–65%) with a 12 h light cycle. Animals were allowed food (Altromin, 1328) and water ad libitum. Animals were randomly assigned to the experimental groups at the time of bleomycin/PBS application for lung injury and for the application of antimiR. After the randomization exercise, PBS applications were carried out prior to bleomycin applications using the micro sprayer or while applying the antimiR using the flexiVent nebulizer unit to facilitate washing the units thoroughly and prevent cross-contamination of reagents between experimental groups. Person carrying out the lung function was blinded for the antimiR applied to the mice. All stainings, histological assessment and scoring were blinded. No randomization or blinding was performed on scRNA-seq experiments for mice cell suspensions used in library generation (cell viability > 85%), processing and data analysis.

### Protein docking

Protein-ligand interaction predictions were made using the Proteins.plus web service (https://proteins.plus). MRC1 receptor structure (CRD4 domain) was downloaded from the RCSB PDB (protein data bank) server and water molecules were removed. On this platform, we performed molecular docking analyses such as protonation (Protoss), binding site detection (DoGSiteScorer), structure ensemble assembly (SIENA) as well as protein-ligand docking (JAMDA). Using SIENA, we were able to identify the appropriate binding site conformation by searching the PDB database for already known binding site conformations.

### Intratracheal administration of bleomycin

10–12 weeks old C57BL/6 N wild-type female mice were administered with bleomycin hydrochloride (Merck Sigma Aldrich Taufkirchen, Germany) diluted in 50 μL sterile PBS at 2 U/kg body weight of mice using a micro sprayer (Penn Century device distributed by BioJane Limited, China). Afterwards, mice were kept in O$_2$-enriched (30% V/V) cages for 14 days according to the recommendation received from the animal regulatory authorities.

### Aerosolized application of oligonucleotide inhibitors

AntimiRs were synthesized as fully phosphorothioated locked nucleic acid (LNA)/deoxyribonucleic acid mixmers (baseclick, Munich, Germany or Axolabs, Kulmbach, Germany). AntimiRs were applied using a nebulizer unit attached to the flexiVent FX2 device (Scireq, Canada), which was operated using the flexiWare v8.1 as described below.

Mice were deeply anaesthetized with an intraperitoneal injection of medetomidin (0.5 mg/kg), midazolam (5 mg/kg) and fentanyl (0.05 mg/kg). Endotracheal intubation was performed using a 22 G, 25 mm intravenous plastic catheter (B. Braun Melsungen AG, Melsungen, Germany), without suture sealing the wall of the trachea around the intubation catheter. The cannula was then attached to a

mechanical ventilator for mice that was equipped with a nebulizer unit (Aeroneb Lab with small droplet diameter 2.5–4 μm, Aerogen Inc., Ireland). Aerosol output rate of 0.24525 mL/min and a delivery ration of 25.406% obtained from routine measurements on this instrument was used for inhalation experiments. The duty cycle of the nebulizer was set to 25% for a duration of 20 s. For each mouse, the nebulizer was filled with 20 μL of 10 mg/mL oligonucleotide inhibitor and was active for 40 ms per breath with a mechanical ventilation at 120 breaths/min, 0.4 mL/kg tidal volume, an inhalation-expiratory ratio of 2:1 and a positive end expiratory pressure (PEEP) at 3 cm $H_2O$. After the procedure, anaesthesia was antagonized with a subcutaneous injection of Flumazenil (0.5 mg/kg) and Atipamezol (2.5 mg/kg).

### Lung function using flexiVent
Mice were anaesthetized with an intraperitoneal injection of 120 mg/kg ketamine (cp-pharma) and 20 mg/kg xylazine (cp-pharma) and placed on a heated table. Orotracheal intubation was performed using an 18 G metal cannula (Scireq, Canada), with a suture sealing the wall of the trachea around the cannula. Animals were then connected to the FX2 device (flexiVent, Scireq, Canada) and mechanically ventilated at a respiratory rate of 120 breaths/min, a tidal volume of 0.4 mL/kg and a PEEP set at 3 cm $H_2O$.

First 'Deep Inflation' protocol was carried out to determine the inspiratory capacity (IC). The deep inflation gradually inflates the lung for 3 s to a pressure of 30 cm $H_2O$ and holds that pressure for another 3 s to allow the alveolar pressure to equilibrate to the applied pressure.

The forced oscillation perturbation Prime-8 was performed. This volume-driven, 8 s long measurement subjects the lung to a standardized test signal of oscillatory frequencies well above and below the subject´s ventilation frequency. Different frequencies will probe different regions of the lung. The measurements were fitted to constant phase model to obtain Newtonian resistance (Rn), tissue damping (G) and tissue elastance (H).

PV loops were generated via the pressure-driven ramp-style pressure volume manoeuvre (PVr-P) to obtain compliance (C) and elastance (E) of the respiratory system, an estimate of inspiratory capacity (A), shape constant (K) and the area enclosed by the PV loop (Area).

### Isolation of primary cells from adult mouse lung
After the lung function measurements were carried out, bronchoalveolar lavage fluid was collected by sequential (about 5 times) instillation and aspiration of PBS containing 2 mM EDTA through insertion of cannula in the trachea while the mice were still under anaesthesia. After the collection of the lavage cells, the lungs were perfused with PBS through the heart after making an incision into the right atrium. Then, the right lung was ligated at the main bronchus and was separated from the left lung that was still connected to the cannula. Right lung lobes were removed and washed in ice-cold PBS containing 2 mM EDTA and were cut into very small pieces in 3 mL enzyme digestion solution containing Collagenase II (Worthington), DispaseII (Merck Sigma Aldrich) and DNaseI (Merck Sigma Aldrich) followed by incubation at 37 °C for 25 min. The digestion was stopped by addition of 500 μL of fetal calf serum (Merck Sigma Aldrich). Cells were washed in FACS buffer (PBS containing 2 mM EDTA and 0.5% bovine serum albumin) and filtered through a 70 μm cell strainer. The left lung lobe was then subsequently filled with 4% paraformaldehyde for histological analyses.

### Flow cytometry/cell sorting (mouse)
Cell pellets were then treated with anti-mouse CD16/CD32 (Fc block, diluted 1:50 in FACS buffer, BD Biosciences at 4 °C for 15 min before being incubated with magnetic microbeads-conjugated anti-CD45 primary antibody (Miltenyi Biotec) at 4 °C for 20 min in a shaker.

Leukocyte fraction was separated from other cell fractions using AutoMACS (Miltenyi Biotec) with the program 'possel'.

Immune cell-enriched fractions from the lung were either stained with antibodies against CD45-FITC (30-F11, 1:100 dilution, BioLegend), F4/80-PECy7 (BM8, 1:100 dilution Thermofisher Scientific, SIGLECF-PerCP-eFluor-710 (1RNM44N, 1:100 dilution, Thermofisher Scientific), CD11b-PE (M1/70, 1:160 dilution, BioLegend) and CD24-PE/Dazzle 594 (M1/69,1:80 dilution, BioLegend) to isolate macrophages, or with antibodies against CD45-FITC (30-F11,1:100 dilution,BioLegend), Ly6G-PE/Dazzle 594 (1A8, 1:80 dilution, BioLegend), CD3-PECy7 (17A2, 1:100 dilution, BioLegend) and CD19-PECy5 (1D3, 1:80 dilution, Thermofisher Scientific) to isolate neutrophils and T cells. The flow through lung cell suspension enriched for non-immune cell fractions were stained with antibodies against CD45-FITC (30-F11,1:100 dilution,BioLegend), CD140a-PECy7 (APA5, 1:100 dilution, Thermofisher Scientific), CD105-PE (MJ7/18, 1:100 dilution, Thermofisher Scientific) and EPCAM-PE/Dazzle 594 (G8.8, 1:100 dilution, BioLegend) to analyse fibroblast, endothelial and epithelial cells, respectively. Anti-CD45 antibody was used in the non-immune cell fractions to exclude leukocytes. Cells were analysed using Sony SH800 sorter using the 130 μm sorting chip.

### Analysis of antimiR-21-FAM in pulmonary cells in vivo
To test the uptake of antimiR in vivo, FAM-labelled unconjugated and carbohydrate-conjugated LNA-antimiR-21 (baseclick) at indicated doses (0.625, 1.25 mg/kg and 2.5 mg/kg) were administered to mice via aerosolized delivery using the flexiVent system. Mouse treated with PBS was included on each experiment day to normalize the background fluorescence and to facilitate comparison across replicate and treatment groups performed on different days. Two hours later, cells were isolated from BALF and lung tissue and the immune-cells were separated with magnetic microbeads-conjugated anti-CD45 primary antibody using AutoMACS. The cells were then stained with antibodies as follows: (1) Immune cell-enriched fractions to assess macrophages: anti-CD45-PECy5 (30-F11, 1:80 dilution, BioLegend), anti-CD206-PE/Dazzle 594 (C068C2, 1:40 dilution, BioLegend), anti-F4/80-PECy7 (BM8, 1:100 dilution, Thermofisher Scientific) and anti-SIGLECF-PerCP-eFlour-710 (1RNM44N, 1:100 dilution, Thermofisher Scientific); (2) Immune cell-enriched fractions to assess neutrophils, B and T cells: anti-CD3-PECy7(17A2, 1:100 dilution, BioLegend), anti-CD11b-PE (M1/70, 1:60 dilution, BioLegend), anti-CD19-PECy5 (1D3, 1:80 dilution, Thermofisher Scientific) and anti-Ly6G-PE/Dazzle 594 (1A8, 1:80 dilution, BioLegend; (3) CD45-negative fraction: anti-CD45-PECy5 (30-F11, 1:80 dilution, BioLegend), anti-CD105-PE (MJ7/18, 1:100 dilution, Thermofisher Scientific), anti-CD140a-PECy7 (APA5, 1:100 dilution, Thermofisher Scientific), anti-E-cadherin-PerCP/Cy5.5 (DECMA-1, 1:20 dilution, BioLegend). Prior to staining of the cells, Zombie Red (BioLegend) diluted in PBS was used to stain dead cells. We then analysed the data using FlowJo software (v10.7.1) to calculate the number of FAM-positive cells and median fluorescence intensity.

### Lysis of cells for RNA sequencing
Sorted cells were centrifuged, and the pellet resuspended in 5 μL containing 1X NEBNext Cell Lysis Buffer (New England Biolabs) and 2000 U/mL RNase inhibitor (New England Biolabs).

### Human precision-cut lung slices (hPCLS) preparation and culturing
hPCLS were prepared as described previously[22]. In short, lung tissue was inflated with 42 °C warm 3% (w/v) low-melting point agarose (Merck Sigma Aldrich) in DMEM/F-12, HEPES medium supplemented with L-Glutamine (Thermofisher Scientific), 1% Pen/Strep (Thermofisher Scientific), 0.1% Amphotericin B (Thermofisher Scientific) and 0.1% fetal bovine serum (Merck Sigma Aldrich) (cultivation medium). The inflation was done by inserting a cannula in several bronchi until the tissue was properly filled. Then, the tissue was incubated for

30–60 min in 4 °C cooled cultivation medium. Afterwards the tissue was sectioned into smaller pieces and a lung section which was properly filled with agarose was mounted on a vibratome holder using a cyanoacrylate glue. The tissue was sliced with a Leica vt1200s vibratome with the following settings (thickness: 400 μm, speed: 0.01 mm/s, temperature: 4 °C, amplitude: 1.4 mm). After cutting, slices were transferred to a 10 cm culture dish filled with cultivation medium and 1 cm punches were cut out. The 1 cm punches were subsequently cultivated in a 24-well TC plate in a 37 °C, 5% CO$_2$ incubator for up to three weeks.

## Digestion of human lungs

After washing, hPCLS were cut into fine pieces in HBSS (Merck Sigma Aldrich) digestion buffer containing Collagenase Type IV 575 U/mL (Worthington), DNAse I 0.3 mg/mL (Merck Sigma Aldrich), Dispase II 2 U/mL (Merck Sigma Aldrich), Elastase 1.5 U/mL (Worthington) and CaCl2 5 mM (Merck Sigma Aldrich), and incubated for 1 h at 37 °C. The cells were filtered through a 40 μm cell strainer and the digestion was stopped with EDTA 50 mM and FBS 10% v/v (Merck Sigma Aldrich) in DPBS. An erythrocyte lysis for 10 min at RT was performed by adding 1 mL Red Blood Lysis buffer (Miltenyi Biotec) to the cells.

## Flow cytometry/cell sorting (human)

Another filter step followed and the centrifuged cells were resuspended in 40 μL FACS buffer (PBS containing 2 mM EDTA and 0.5% bovine serum albumin) containing 10 μL of FcR Blocking Reagent (Miltenyi Biotec). The cells were blocked for 10 min at 4 °C. Then, the cells were incubated with CD45 magnetic beads (Miltenyi Biotec) for 15 min at 4 °C and the leukocyte fraction was separated from other cell fractions using AutoMACS (Miltenyi Biotec) with the program 'possel'. Immune cell-enriched fractions from the lung were stained with Zombie Violet dye (BioLegend) and antibodies against CD45-FITC (HI30, 1:20 dilution, BioLegend) and CD206-PECy5 (15-2, 1:20 dilution, Sony Biotechnology) to isolate macrophages. Cells were analysed using Sony SH800 sorter using the 130 μm sorting chip.

## Analysis of antimiR-21-FAM in hPCLS

100 μL of FAM-labelled RCS-21 (concentration 6.25 μM) was added to each human lung slice in a 24-well plate. After 2 h 3 × 8 hPCLS were pooled and digested (see digestion of human lungs). Then, cells were prepared for FACS analysis (see Flow cytometry/cell sorting (human)). CD45+ fractions and CD45- negative fractions were stained with CD45-PE (HI30, 1:20 dilution, BioLegend) and CD206 PE-Cy5 (15-2, 1:20 dilution, Sony Biotechnology) for 30 min at 4°. After one washing step with PBS, cells were analysed using Sony SH800 sorter with a 130 μm sorting chip and FlowJo software (v10.7.1).

## Treatment of hPCLS with SARS-CoV-2 and RCS-21

SARS-CoV-2 infection of hPCLS was carried out in collaboration with the S3 lab (AG Protzer) from the Technical University of Munich. The slices were kept in culture medium until the day of infection. Thus, the slices were washed 1x with PBS, followed by the infection with omicron B1.1.529 (7808190) in 150 μL culture medium at a multiplicity of infection of 1 (MOI1). 4 h later, 300 μL of culture medium was added and incubated for 20 more hours. Afterwards, the slices were washed 1x with PBS and RCS-21 in a concentration of 25 μM in a total volume of 100 μL was added. 4 h later 300 μL of culture medium was added and incubated for 68 more hours. At the end, the slices were washed three times with PBS and either fixed with 4% PFA for 24 h for immunostaining or lysed in TRIzol for RNA isolation.

## RNA isolation

hPCLS or frozen tissues were homogenized in TRIzol (Thermofisher Scientific) using an Ultra-Turrax®. RNA isolation was performed

according to manufacturer instructions. TapeStation 4200 (Agilent) was used for quality control.

## PolyA-RNA sequencing

70–100 ng of extracted RNA or sorted cells in lysis buffer were used to generate a polyA-enriched library using the NEBNext® Single Cell/Low Input RNA Library Prep Kit for Illumina® (NewEngland Biolabs). In short, RNA was reverse transcribed using an oligo d(T)-containing RT primer, followed by a PCR amplification. DNA Fragmentation, End Repair and dA-Tailing were acquired in one step and adaptors with a single T overhang were ligated to the processed DNA fragments. To obtain the DNA fragments of interest, size selections using magnetic beads were performed. The last step involved the PCR amplification of the final library, where barcodes were introduced to each library. Barcoded libraries were then cleaned up and pooled for sequencing for a run on a NextSeqHigh (75 or 100 bp, paired end). An average of 40 Mio reads per sample was achieved.

For analysis, template-switching oligo-related sequences and Illumina adapters from raw reads were trimmed with Flexbar 3.5.0[23]. In the case of the human lung tissue and hPCLS, the trimmed raw reads were aligned to the human reference genome hg38 using the Galaxy Europe tool HISAT2 2.1.0[24] and for transcript assembly and quantification StringTie 2.1.1[25] was used. Downstream analysis for differentially expressed genes was performed in R. Read counts (>1) were normalised and RUVSeq 1.26.0[26] was used to remove unwanted variation between samples (k = 4–6). For a differential expression analysis edgeR 3.32.1[27] was performed. For the SARS-CoV-2 alignment, the trimmed raw reads were first aligned to human reference genome hg38 using the Galaxy Europe tool HISAT2 2.1.0. Then, unaligned reads using BAM filter 0.5.9 were subtracted and were aligned to the SARS-CoV-2 (GCF_009858895.2) reference genome.

For sorted mouse lung cells, STAR 2.7.8a[28] was used for alignment to mouse reference genome 10 (mm10). StringTie merge (v 2.1.1) was used to merge transcripts and DESeq2 1.32.0[29] was used for differentially expressed analysis.

For the prediction of miR-21 conserved targets, TargetScanMouse 7.2. and TargetScanHuman 7.2. were used[30].

For visualisation of the results ggplot2 v 3.3.6 was used.

## Small RNA sequencing

100–200 ng of extracted RNA were used to generate small RNA libraries using the TrueQuant SmallRNA Seq Kit for Ultra Low input® (GenXPro, Frankfurt, Germany) or NEBNext® Small RNA Library Prep Set for Illumina® (New England Biolabs, Frankfurt, Germany). In brief, adapters were ligated to the 3' and 5' end of the small RNAs. The products were then reversed transcribed and amplified by PCR. Instead of running a gel electrophoresis as it was the case for the NEBNext kit, in the GenXPro kit a double-sided size selection was performed to enrich for small RNAs. The individual libraries were pooled and sequenced on a MiSeq or NextSeq500 (50 bp, single-end). For analysis, raw reads were trimmed within the Galaxy Europe Server to remove adaptors and UMIs using Trim Galore! 0.6.7. or Cutadapt 3.4 and fastp 0.23.2. After trimming, filtering for low quality reads (Phred score < 20) and removal of reads smaller than 18 bp was performed. MirDeep2 2.0.0[31]. was used for identification and quantification of microRNAs between samples. The reads were mapped to the human reference genome hg38 for human samples or to mm10 for mouse samples and, reference files of hairpin and mature miR-sequences from miRbase 22.1 were provided for better prediction.

## Single-cell RNA sequencing

Lungs were harvested from RCS-21 and RCS-control-treated wild-type mice 14 days after bleomycin injury and lungs were digested using Collagenase II/ DispaseII/ DNaseI enzyme cocktail at 37 °C for 25 min. Digestion was stopped with 500 μL FCS as described above. Lung cell

suspension was then incubated with rat anti-mouse CD16/CD32 (BD Biosciences) at 4 °C for 20 min, anti-CD45 microbeads (Miltenyi Biotec) at 4 °C for 30 min and were separated into leukocytes (CD45-enriched) and non-leukocytes using AutoMACS. Isolated leukocyte and non-leukocyte fractions were filtered through a 40 μm Nylon cell strainer (Falcon) and counted using Invitrogen Cell Countess (ThermoFisher Scientific) and Trypan Blue staining (ThermoFisher Scientific). Leukocyte and non-leukocyte fractions from each mouse were mixed in the ratio of 1:1 to amplify the expression profiles of non-leukocyte cells. This practice of normalisation obscures the changes in cell numbers between different treatment conditions. Cells isolated from three mice from each experimental group were individually incubated with Total Seq A hashing antibodies H2, H3 and H4 (BioLegend), respectively, at 4 °C for 30 min for multiplexing. After washing three times with PBS, cells were resuspended in PBS at 1500 cells/μL and cells from three mice of each experimental group were pooled together (cell viability > 85%). 15,000 cells were loaded into a single channel of the Chromium system (10X Genomics) for each of the experimental group, and the 3′-gene expression (3′-GEX) libraries were prepared according to manufacturer's instructions (10X genomics) up until the cDNA amplification step. Amplification of cDNA was carried out after spiking in 0.2 μM each of hashtag antibody oligonucleotide-derived tag (HTO) additive primer according to instructions in 10X Genomics manual. Following the amplification step, 0.6X SPRIselect reagent was added to separate the cell hashtag-containing fraction in the supernatant from the larger cellular mRNA-derived cDNA fraction bound to the beads. The cellular mRNA-derived cDNAs were processed according to the standard 10X Genomics protocol to generate the mRNA library. The HTO supernatant was further purified to remove excess single-stranded oligonucleotides by adding 1.4x SPRIselect reagent. Purified HTO supernatant was then amplified using SI-PCR and TruSeq i7 index primers (Sham control oligo D701_s, Sham RCS-21 D702_s, Bleo control oligo D704_s, Bleo RCS-21 D705_s) and then was purified using SPRIselect reagent. 3′-GEX:HTO (14.:1) libraries of all samples were pooled together and sequenced on a NovaSeq 6000 S1 chip (R1/28/i7/8/i5/0/R2/91) to obtain about 3000-5000 reads per cell for HTO library and 30,000 reads per cell for 3′-GEX library.

Cell ranger (v 6.0.0, 10X Genomics) was used to process raw sequencing data and Seurat (v 4.1.0)[32] was used for downstream analysis. Sequencing reads were aligned to mouse reference genome build 39 (GRCm39; vM27). Cells with < 200 genes or > 3000 genes or mitochondrial genes greater than 5% were filtered out. Principal components that contributed to the dimensionality of the data were identified using a Jack straw test and used for unsupervised graph-based clustering (resolution 0.3) and Uniform Manifold Approximation and Projection (UMAP) for embedding and visualization. Individual objects were merged together using the merge function as there were no condition-specific differences in the clustering of cells were observed. FindMarkers tool within the Seurat package was then used to identify individual marker genes for each cluster. FindMarkers with wilcox test (Seurat) was used to identify differentially expressed genes between two clusters (log2 fold change > 0.25 or log2 fold change < −0.25, and adjusted $p$ value < = 0.05). We calculated enrichment module scores for pathway gene sets using Seurat's function "AddModuleScore". The following marker genes were used to for the macrophage metabolism: Glycolysis: *Slc2a1*, *Hk1*, *Hk2*, *Pgk1*, *Eno1*, *Pkm*, *Pfkfb3*, *Pgam1*, *Tpi1* and *Pfkl*; TCA: *Me2*, *Idh1*, *Ogdh*, *Idh2*, *Mdh1*, *Sdha*, *Aco1* and *Idh3g*; Arginine: *Arg1*, *Arg2*, *Nos1*, *Nos2*, *Oat*, *Asl* and *Ass1*; and fatty acid ß-oxidation: *Cpt1a*, *Cpt2*, *Fabp1*, *Akt2*,*Cd36*, *Acadvl* and *Acadm*.

Processed data of publicly available scSeq datasets ("GSE141259"[33]; DUOS-000130 [https://duos.broadinstitute.org (study ID DUOS-000130)][3]) were obtained and were re-clustered again as described above. Since condition-specific clustering of cells were observed, individual objects were combined together using the RunFastMNN algorithm that scales the values of the objects relative to

each other and cluster the cells in the same manner. The cell type annotations as provided by the authors in the meta data were used.

## Gene ontology analysis

Gene ontology enrichment analysis for mouse samples was performed on the top 400 deregulated genes using DAVID 6.8[34]. A hypergeometric test (Fisher's exact test) was used to calculate the statistical significance of gene overrepresentation followed by a Bonferroni correction for multiple testing to estimate proportion of enriched genes that may occur by chance for the given set of genes. The top significantly enriched GO terms were then analysed to reduce the number of redundant GO terms and visualised using the GOPlot R package 1.0.2.

In the case of human samples, mitochondrial genes and ribosomal genes were removed and gene ontology were performed on genes with $\log_2FC < −1$ & $\log_2FC > 1$.

## Sirius red/fast green

For the analysis of collagen deposition, 8 μm paraffin sections of lung tissue were stained with Sirius red and Fast green. Whole lung images were taken with a 10X objective using an AxioObserver.Z1 (Zeiss, Jena, Germany) motorised scanning-stage microscopy and the images were analysed using MetaMorph software v 7.10.1.161 (Molecular Devices, Downingtown, USA).

## MicroRNAscope-immunofluorescence co-staining

According to manufacturer instructions (ACD Bio-Techne), formalin-fixed paraffin-embedded tissue slices from COVID-19 patients and control patients were deparaffinized with xylene and ethanol, followed by a post-fixation with 12% formaldehyde for two hours. Slices were then treated with hydrogen peroxide for 10 minutes. A manual target retrieval for fifteen minutes (99 °C) followed. Next, the slices were treated with Protease III for 30 min, 40 °C. In case of hPCLS, slices were post-fixed in 4% PFA for two hours at 4 °C and dehydrated via an ethanol concentration row. Then, slices were blocked with hydrogen peroxide and permeabilized with Protease IV solution for 30 min at room temperature. In both cases, miRNAscope Probe - SR-hsa-miR-21-5p-S1 (Cat. #728561-S1) was then added to the slices and incubated in a humidity control chamber for two hours. The signal amplification reagents 1-6 were added sequentially and incubated for either 15 or 30 minutes. Afterwards, samples were treated with detection reagents (Fast Red-B and Fast Red-A in a 1:60 ratio) and in case of lung tissue, a co-immunofluorescence staining was performed. The slices were permeabilized in 0.15 % Triton X buffer and blocked in 4% NGS for 1 h at room temperature. Next, they were stained with mouse anti-human CD68 (KP1, 1:100 dilution in PBS, Thermofisher Scientific), a marker for macrophages and incubated with Alexa Fluor 488 goat anti-mouse (1:100, Thermofisher Scientific) for 1 h at room temperature. Incubation with DAPI (2 μM, Abcam) followed. For imaging, the samples were mounted with Vectashield Hardset antifade mounting medium with DAPI (Vector Laboratories) and protected with a cover slip. A Leica SP5 (Leica) confocal microscopy was used for imaging and ImageJ 1.53n was used for analysis. Segmentation analysis for identification of positively stained cells and nuclei were performed using MetaMorph software (v 7.10.1.161).

Mice lung cryosections (5 μm) were fixed with 4% paraformaldehyde (PFA) and stained with rat anti-mouse CD68 (FA-11, 1:50 dilution in 3% goat serum/PBS, BIO-RAD) or rabbit anti-mouse MRC1 (1:100 dilution in 3% goat serum/PBS, abcam) at 4 °C overnight. The sections were washed with 0.05% (V/V) Tween20/PBS (PBST) and were stained with AlexaFluor 647-conjugated goat anti-rat antibody (1:100 dilution in PBST, Thermofisher Scientific) or AlexaFluor 647-conjugated anti-rabbit antibody (1:100 diluted in PBST, ThermoFisher Scientific) at 37 °C for 3 h. Nuclei were stained with DAPI (abcam) or SYTOX green. Images were acquired with Leica SP5 confocal microscope with a 20X

objective and at least 4000 nuclei per mice were analysed using MetaMorph software v 7.10.1.161 (Molecular Devices).

## Quantitative real-time PCR

The expression of miR-21a-5p was quantified using LNA-enhanced microRNA assays (Exiqon). 10 ng of total RNA were reverse-transcribed using the Universal cDNA Synthesis kit II (Qiagen). The cDNAs were then quantified using the Fast Start Universal SYBR Green master mix (Roche) and LNA-enhanced miRCURY PCR primer sets for miRNAs (Qiagen). For real-time PCRs, thermal-cycling parameters were applied as recommended by the manufacturer. U6 snRNA was used as normalisation control.

## Statistics

Data denote mean and individual values. Statistical analysis was performed using Graphpad Prism software package (version 8.4.2). Data distribution was assessed by the Shapiro–Wilk test for normality. The Bartlett test was performed to test the homogeneity of variance, and the Spearman rank correlation test was performed to test heteroscedasticity. For data sets that did not pass the heteroscedasticity test, values were log-transformed before statistical testing. Differences among multiple means were assessed either by 1-way or 2-way ANOVA followed by the Sidak test. To compare only two groups, unpaired $t$-test was performed. Two-sided Kolmogorov-Smirnov test was used for testing significance between cumulative distribution curves. Asymmetric nonlinear regression analysis used for curve fitting and comparison of the fits for significance using 3 parameters (EC50, Hillslope, Ymax). A $p$ value of $< 0.05$ was considered significant.

## Reporting summary

Further information on research design is available in the Nature Portfolio Reporting Summary linked to this article.

## Data availability

All raw and processed NGS datasets generated in this study have been deposited to NCBI GEO under the accession number GSE235136 and are available under the following link. All source data is provided with this paper. The authors declare that all data that support the findings of this study are available with the article and its Supplementary Data. Source data are provided with this paper.

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

## Acknowledgements

The authors thank the following members of the Institute of Pharmacology and Toxicology: J. Auerswald for primary cell isolations, mouse surgery and lung function measurements; S. Brummer for lung histology; A. Welling, V. Philippi and C. Klug for supervision of experimental animal studies; J. Decker and L. Koblitz for RNA library generation for sequencing; H. Schiller (Institute of Lung Health and Immunity, Comprehensive Pneumology Center Munich) for advice on the bleomycin model; V. Benes (Gene Core facility, EMBL, Heidelberg), E. Graf and B. Riccardo (NGS Core Facility, Helmholtz Center Munich) and Rupert Öllinger (Institute of Molecular Oncology and Functional Genomics, Translatum Cancer Center, School of Medicine, Technical University of Munich) for RNA sequencing. We thank the team at the Institute of Legal Medicine, Ludwig-Maximilians-Universität (LMU) München and the Institute of Legal Medicine, University Medical Center Hamburg-Eppendorf for assisting in the collection of lung tissues and K. Püschel (Institute of Legal Medicine, University Medical Center Hamburg-Eppendorf) for advice. We thank M. Lütgehetmann and S. Reuchert (Institute of Medical Microbiology, Virology and Hygiene, University Medical Center Hamburg-Eppendorf) for their contributions to the initial RNA isolations from human lungs; A. Herrmann (Institute of Virology, Helmholtz Munich, Technical University of Munich (TUM), School of Medicine) for his support in carrying out the hPCLS experiments at the S3 facility. Images used in the schemes (Fig. 1a - human cohort and lung; Fig. 3a - mouse, lung and nebulizer; Fig. 5a - human silhouette and virus particles and Fig. S11a - lung) and human silhouette (Figs. 1c–e, 2e, 5a, S1b, S5b, S17) were created with BioRender.com. The authors acknowledge the support of the Freiburg Galaxy Team: Björn Grüning, Bioinformatics, University of Freiburg (Germany) funded by the Collaborative Research Centre 992 Medical Epigenetics (DFG grant SFB 992/1 2012) and the German Federal Ministry of Education and Research BMBF grant 031 A538A de.NBI-RBC. This study was supported by Deutsche Forschungsgemeinschaft (DFG, project ID 403584255–TRR267, S.E.), Bayerische Forschungsstiftung through CoVmiR (AZ-1452-20C, S.E.), BMBF in the framework of the Cluster4future program (CNATM - Cluster for Nucleic Acid Therapeutics Munich, S.E.) and European Union (ERA-CVD, MacroERA [01KL1706], S.E.). Furthermore, this work was supported by DFG through ResearchTraining Group GRK2338 (P09, S.E. and D.R.) and the European Union's Horizon 2020 research and innovation programme under the Marie Skłodowska-Curie grant agreement No 813716 (TRAIN-Heart, S.E. and D.R.).

## Author contributions

C.B., D.R. and S.E. conceived the project, designed experiments and wrote the manuscript. S.E. supervised the overall project and the data analysis. C.B. and D.R. contributed equally, led the project, designed protocols, conducted experiments, performed data analyses and interpreted the data. P.V. generated small RNA and transcriptome datasets of different lung cell types isolated from healthy mice or human lungs and small RNA sequencing of lung tissue after bleomycin injury. F.W. contributed to miRNAscope stainings, imaging and ImageJ analyses. C.B., M.F. and C.C.C. carried out the experiments at S3 facility. U.P. supervised the experiments using SARS-CoV-2 at the S3 facility. A.B. performed the pulmonary injury model, lung function testing and inhalation using the flexiVent device. T.A. helped with collecting lung samples and preparing hPCLS. J.S., J.P.S., M.G., S.S. and H.H oversaw and performed tissue collection and provided clinical advice. C.S.W. provided materials and technical advice on lungs and R.R. carried out small RNA sequencing and advice on macrophage experiments. T.F. contributed to the design of RCS-21 and supervised oligonucleotide synthesis. All authors reviewed and approved the manuscript.

## Funding

## Competing interests

Technical University of Munich has filed an intellectual property right on the therapeutic use of mannose-coupled antimiR-21 with D.R. and S.E. named as inventors. S.E. and T.F. are founders of RNATICS GmbH, a biotech company focussed on macrophage RNA therapeutics. After the completion of this study, D.R. joined RNATICS GmbH. Other authors do not have any conflicts.
