## [Peer Review File · Nature Communications]

Trimannose-coupled antimiR-21 for macrophage-targeted inhalation treatment of acute inflammatory lung damageREVIEWER COMMENTS

Reviewer #1 (Remarks to the Author):

This is a very interesting manuscript that reports two important findings. First, that chemically linking an anti-miR oligoribonucleotide to a trimannose sugar promotes its specific targeting to lung macrophages. Second, that an anti-miR against endogenous miR-21 modified in such a manner can be used as an inhalator treatment against inflammatory lung damage, with a potential application against COVID-19 disease. Both conclusions are interesting and generally well supported by the data presented. Some suggestions for improvement are the following.

The authors selected trimannose as an Mrc1 ligand. It is unclear why they specifically selected this type of sugar and what its specificity for Mrc1 targeting is. What is the affinity of trimannose to Mrc1? Are there studies on other sugar formulations? Other information on this biochemical interaction?

Fig 3b shows uptake of RCS-21 in pulmonary macrophages. But these experiments are performed at 2 hr after inhalation, when presumably most of the signal is in endocytic vesicles. What is the kinetics of release of the conjugated oligos from endosomes? Is this kinetics different from that of GalNac-conjugates?

The authors convincingly show efficacy of RCS-21 in reversing pathological activation of macrophages and prevention pulmonary dysfunction and fibrosis. But it would be important to show some data on the pharmacological properties of RCS-21 in vivo. How long does the drug persist? What is the level of miR-21 downregulation over time in different lung cell types? Safety profile? Transcriptome analysis at day 14 seems to indicate upregulation of miR-21 targets, but it remains unclear whether this is due to the actual persistence of the oligonucleotide or of the prolongation of its effect.

The experiments shown in Fig 5 using an elegant ex vivo SARS-CoV-2 infection method of primary human lung tissue are quite compelling in terms of efficacy using the analysed surrogate endpoints. However, the observation time is inevitably quite short (5 days) and

one wonders whether there is an effect of treatment on SARS-CoV-2 replication itself, in addition to that on the macrophage response. Does anti-miR21 treatment affect acute infection by the virus in cultured lung epithelial cells, in the absence of macrophages?

Reviewer #2 (Remarks to the Author):

Summary:

The authors sought to develop a novel macrophage-specific targeted microRNA drug that alters lung inflammation. They observed miR-21 was upregulated in whole lung tissue 14 days after bleomycin challenge and increased in whole tissue from patients who died from SARS-CoV-2. They discover that miR-21 is highly expressed in lung macrophages (both alveolar and interstitial) and created a macrophage specific (CXCR3-Cre) miR-21 mouse that is protected from developing bleomycin induced pulmonary fibrosis. They decide to generate a macrophage-targeted carbohydrate-coupled antimir to MRC1 (called RSC-21) and demonstrated it's efficacy using in vivo aerosolized delivery system. Bleomycin challenged WT mice treated with RSC-21 had reduced lung stiffness and decreased collagen staining. In addition, the authors used a human PCLS system to show that RSC-21 reduced inflammation following SARS-CoV-2 infection.

Major:

1. The authors compare an acute single dose of bleomycin in a mouse with humans who died from SARS-CoV-2 infection, but it is not clear what they are examining using these two different models. Are the authors suggesting that SARS-CoV-2 causes fibrosis is a similar mechanism to bleomycin in a mouse or was there another connection between these two lung injury models. Additional rationale needs to be provided to examine how this microRNA is being regulated in these two different model systems.
2. There are several populations of macrophages present within the lung microenvironment. While Figure 1E shows the presence of these cells by staining with the macrophage marker CD68, it's not clear if there are differences in the total number, type of macrophage or activation status of these cells in their different models. In the supplemental figures they distinguish interstitial macrophages from alveolar macrophages via transcriptomics, but the

field has moved beyond this to include ontogeny (monocyte derived vs. tissue resident) as a major contributor of macrophage function and activation. A detailed explanation of the role of miR-21 during different macrophage subpopulations as well as during injury/inflammation settings are needed.

3. In general, there was a lack of inflammation measured in this model. Although the investigators measured differences in other immune cells via scRNAseq and transcriptomics, there was no mention how this oligo treatment altered the number or activation of myeloid cells or other immune cells (such as neutrophils). Additional information regarding a target of how miR-21 is causing these effects in macrophages is needed.

4. The authors demonstrate that there is reduced lung elastance and compliance, decreased collagen accumulation and reduced expression of ECM related genes in the fibroblast cluster with treatment of RCS-21; however, there is no mechanism offered to explain how targeting macrophages could regulate fibrosis or collagen accumulation or expression.

5. The authors make the point that miR-21 is upregulated specifically in macrophages, and that by generating a macrophage-targeted antimir-21 (RCS-21) this alters the development of lung fibrosis; however, in their PCLS culture they do not measure localization of miR-21 or presence of macrophages in their cultures. Yet, they observe significant differences in transcriptomics after infection with SARS-CoV-2. Is this due to RSC-21 affecting macrophages or other cellular populations? The relative role and possible downstream signaling pathways need to further explored.

Minor:

1. The histological section provided within the manuscript only show a very small section, larger sections of the entire lung or lobe should be shown in addition.

2. There was no description regarding the sex of the mice utilized in this study. Additional information should be provided especially since there were relatively small sample sizes (n=3) included in the majority of the studies.

3. It is not clear why the author chose to keep the bleomycin challenged mice in an 'O₂-enriched' cage for 14 days. Were the saline challenged mice also kept in these conditions? Has this been shown to alter fibrosis or recovery of the mice?

4. Line 230-232 is an overstatement, and needs to be reworded

Trimannose-coupled anti-miR-21 for macrophage-targeted inhalation treatment of acute inflammatory lung damage

Response to reviewers:

We thank all the reviewers for their insightful and helpful comments. We have addressed all the questions raised by the reviewers and did so with new experiments, wherever applicable.

Comments from reviewer 1:

- 1. The authors selected trimannose as an Mrc1 ligand. It is unclear why they specifically selected this type of sugar and what its specificity for Mrc1 targeting is. What is the affinity of trimannose to Mrc1? Are there studies on other sugar formulations? Other information on this biochemical interaction?**

We addressed these important questions by providing now considerably more detail on how and why trimannose was selected as a candidate carbohydrate ligand to target macrophages through binding to MRC1 (Mannose receptor 1). Taken together, we started the design of a carbohydrate ligand for this receptor with mannose as the sugar moiety based on published data on ligands for MRC1^{1,2}. It is well known that C-type lectins like MRC1 use complex carbohydrate structures e.g. to detect various pathogens and higher, non-monomeric carbohydrates have been reported to often exert higher binding affinities compared to their monomeric forms.¹ We therefore conducted protein docking analyses (ProteinsPlus platform) to predict binding of different forms of mannose. These analyses yielded higher binding affinity to MRC1 for trimannose in comparison to mono- and dimannose. In addition, branched trimannose yielded higher affinity compared to linear trimannose. We did not explore for even more complex carbohydrate ligands to facilitate synthesis, limit size and prevent steric interference with receptor-mediated internalisation. In the revised manuscript, we are now presenting our selection strategy for trimannose as a suitable ligand to target MRC1-expressing cells in considerably more detail as previously, including the data from our docking study. These data are presented **in new Supplementary Figure S8, the revised results section on page 6 (lines 13-18) and the revised methods section (page 13, lines 16-22).**

With regard to the specificity of the branched trimannose (KD 4.62 μ M for MRC1 compared to monomannose KD > 100 μ M)³ chosen in this study, we are not aware of any published data regarding the many other candidate receptors that have been reported to bind various forms of mannose. The following arguments let it appear plausible, that the vast majority of trimannose-coupled RCS-21 is bound and taken up by MRC-1 as compared to other receptors: (i) inhaled RCS-21 is calculated to reach very high concentrations at the target cells (in the low mM range in mice) and must be expected to completely saturate the MRC1 receptor population (estimated concentration approx. 1000-fold over KD) and likewise other (less expressed) receptors. (ii) Given the exceptionally high expression of MRC1 on our target cell population compared to other carbohydrate binding receptors and (iii) it's well documented high capacity of MRC1-mediated uptake² we consider MRC1-mediated uptake as the bona fide dominating route for uptake of RCS-21.

We are now addressing this point **in the new Supplementary figure S6 and changed the result section on page 6, lines 5-7 in the revised manuscript accordingly.**

2. **Fig 3b shows uptake of RCS-21 in pulmonary macrophages. But these experiments are performed at 2 hr after inhalation, when presumably most of the signal is in endocytic vesicles. What is the kinetics of release of the conjugated oligos from endosomes? Is this kinetics different from that of GalNAc-conjugates?**

We agree with the reviewer that most of the drug substance locates to the endosomal compartment as has been reported for other oligonucleotide drugs and in particular so for GalNAc-coupled siRNA⁴. The latter has been shown to strongly accumulate in endosomes with very prolonged release kinetics up to many weeks⁵. This depot effect together with the exceedingly slow release of GalNAc-conjugated siRNA is sufficient for efficient and sustained target knockdown, thus ensuring a long-lasting effect of the GalNAc-siRNA⁵. In fact, the prevailing interpretation of the scientists behind the currently leading drug program using GalNAc-technology propose that the trapping-like enrichment of GalNAc-siRNA conjugates in endosomes does not preclude, but rather enables, their favourable pharmacokinetic profile. While at present, we do not have a detailed assessment of the endosomal release kinetics of RCS-21, we have found pronounced de-repression of the miR-21 targetome until 10 days after application. While this is already a rather long duration, we presume that the total duration of action is even longer. We kindly make the reviewer aware that a quantitative, time-resolved determination of endosomal release kinetics would require a rather extensive and technically challenging set of experimentation and should be the subject of a future study. The Alnylam team only recently published the above referenced data after a > 20 years drug development program on their GalNAc-technology⁶. In response to this reviewer and to clarify this important point to **the reader, we have changed the results section of the revised manuscript (Page 8, lines 20-25) accordingly.**

3. **The authors convincingly show efficacy of RCS-21 in reversing pathological activation of macrophages and prevention pulmonary dysfunction and fibrosis. But it would be important to show some data on the pharmacological properties of RCS-21 in vivo. How long does the drug persist? What is the level of miR-21 downregulation over time in different lung cell types? Safety profile? Transcriptome analysis at day 14 seems to indicate upregulation of miR-21 targets, but it remains unclear whether this is due to the actual persistence of the oligonucleotide or of the prolongation of its effect.**

These are indeed important points raised by this reviewer, and we have conducted a number of new experiments to address them. As to the pharmacological properties, we have in the meantime obtained the plasma half-life of RCS-21 in rats as part of the current IND-enabling studies. After i.v. application, the estimated plasma half-life was approx. 20 min, which is in good agreement with the short plasma half-lives that have been determined for the GalNAc-conjugates⁷. Once we have obtained a full pharmacokinetic profile of RCS-21 (ongoing at present, expected to be completed by the end of 2023), we plan to report this as an independent publication. In response to this reviewer, we are now including the already obtained information on the plasma half-life of RCS-21 **in the revised results section of the manuscript on page 7, lines 9-11.**

With regard to the level of miR-21 downregulation in different lung cell types, we have conducted a new experiment, where we i) measured miR-21 levels in alveolar cells obtained by lavage (i.e. BALF) 10 days after inhalation of RCS-21 to mice (2.5 mg/kg). Here, we found a highly significant reduction of miR-21 (-58% Bleo RCS-21 compared to Bleo control oligo). We are now reporting these new data **in the new**

Supplementary Figure S10b of the revised manuscript and changed the results part accordingly (page 7, lines 20-22). While we cannot provide a timeline of miR-21 expression analysis in different cell types, we were able to analyse mRNA transcriptomes of non-myeloid primary pulmonary cell types, namely endothelial cells, epithelial cells and fibroblasts, isolated from RCS-21-treated mice (likewise 10 days after a single dose of inhaled RCS-21 at 2.5 mg/kg). Inhibition of miR-21 activity (i.e., de-repression of the miR-21 targetome) was absent in endothelial cells, epithelial cells and fibroblasts, but selectively present in macrophages, supporting MRC1-mediated delivery of RCS-21. These new data are now presented in the **revised Supplementary Fig S12e and we changed the results part accordingly (page 8, lines 20-25).**

With regard to the safety profile, mice were examined closely throughout the 14-day course of bleomycin experiments for disease and drug-related side effects. There was no drug-related mortality or morbidity observed in the bleomycin or PBS groups treated with RCS-21. Moreover, histological analysis revealed no significant differences between RCS-21-treated mice and control oligo-treated mice in liver, kidney, and spleen harvested at the endpoint. We further tested the safety profile of RCS-21 systemically by conducting new experiments, where 25 mg/kg of RCS-21 were applied intravenously to rats and for seven consecutive days (10-fold higher than the inhalation dose used in this study). In general, the safety profiles of RCS-21 were found to be favourable, with no indications of organ toxicities, including the liver and kidney. Given that RCS-21 is applied by inhalation and at a dose 10-times lower than the systemic dose, the effects of RCS-21 should be limited to lung tissue, the target organ, with little risk of adverse systemic effects. Of note, antimiR-21 (in its unconjugated form) has been previously applied to extensive toxicology testing in animal models and has been systemically applied to humans in phase 1 (Clinicaltrials.gov; NCT03373786) and lately phase 2 clinical trials (Clinicaltrials.gov; NCT02855268). Collectively, these studies showed very good tolerability⁸. We are reporting the absence of drug-related mortality or morbidity now in the revised results section on **page 7, lines 25-27**. Once the IND-enabling toxicology studies (ongoing at present, including a large animal species) are completed, we plan to report this as an independent publication.

As to the last point within 3) that refers to drug persistence vs. effect duration, we kindly refer the reviewer to our response to **point 2 above**.

- 4. The experiments shown in Fig 5 using an elegant ex vivo SARS-CoV-2 infection method of primary human lung tissue are quite compelling in terms of efficacy using the analysed surrogate endpoints. However, the observation time is inevitably quite short (5 days) and one wonders whether there is an effect of treatment on SARS-CoV-2 replication itself, in addition to that on the macrophage response. Does anti-miR21 treatment affect acute infection by the virus in cultured lung epithelial cells, in the absence of macrophages?**

Thank you for pointing this out. We tested for viral reads in RCS-21-treated hPCLS using RNA-Seq and compared it to infected control slices. We found infection of hPCLS with SARS-CoV-2 to result in a strong transcriptome signature specific for the respective viral RNA reads (see Fig. 5b). Treatment with RCS-21 had no significant effect on these reads, which must be interpreted as the absence of an effect on viral replication. We have included these new data in **new Supplementary Figure S18 of the revised manuscript and changed the text accordingly (Page 10, lines 2-5)**. As there is no effect of RCS-21 on the entirety of the cell populations in human lung tissue, we refrained from the separated analysis of isolated epithelial cells.

In addition, and in line with the above conclusion, our revised manuscript now provides additional evidence that RCS-21 is preferentially taken up by MRC1+ cells as opposed to other pulmonary cells types. We have included these new data in **new Supplementary Figure S16 of the revised manuscript and changed the text accordingly (pages 9 lines 10-12 in the results section and page 18 lines 10-16 in the methods section).**

Comments from reviewer 2:

- 1. The authors compare an acute single dose of bleomycin in a mouse with humans who died from SARS-CoV-2 infection, but it is not clear what they are examining using these two different models. Are the authors suggesting that SARS-CoV-2 causes fibrosis is a similar mechanism to bleomycin in a mouse or was there another connection between these two lung injury models. Additional rationale needs to be provided to examine how this microRNA is being regulated in these two different model systems.**

Thank you for this important point, that was apparently not made clear enough. In short: Yes, we regard bleomycin-induced acute lung damage as a suitable model to study important aspects of acute lung damage and at the time of the study, we regarded it superior to the available transgenic models in that the latter did not show the extent of severe lung damage and the hyperinflammation response that we aim to interfere with the inhibition of miR-21 targeted macrophages. In severe COVID-19, a considerable number of patients develop ARDS leading to fatal airway damage.⁹ Several aspects of this late stage lung damage are likewise seen in bleomycin-induced lung injury^{10,11}. Importantly, activation of macrophages is implicated in both models of tissue damage contributing to disease severity^{12,13,14}. Of note, bleomycin-induced lung injury has also been employed before to study lung injury caused by SARS-CoV-2^{15,16}.

Additionally, and important for this study, the bleomycin model is a well-established model of pulmonary fibrosis. As observed for COVID19-induced fibrosis in human¹⁷, inhalation of bleomycin induces cell death that causes massive inflammation and the latter has been clearly shown to then trigger fibrosis¹⁰. So, while we fully agree with the reviewer that the use of the bleomycin model needs better explanation, we are convinced that our data obtained in this model are valid and relevant. In response to this reviewer, we have now included **new Supplementary Fig. S3 to make the similarities between bleomycin-induced lung injury and COVID-19 more clear and revised the results part accordingly (page 4, lines 22-24).**

With regard to the last point i.e. the regulation of miR-21 in the murine bleomycin model in comparison to the lungs of humans with COVID-19, the commonalities are striking: i) both murine and human lung tissue is the single strongest upregulated microRNA among the microRNAs with expression levels commonly considered relevant (here >1000 normalized reads were used as cut-off) ii) miR-21 is the top1 expressed microRNA in macrophages in both, iii) macrophages in addition strongly accumulate in both models. iv) there is also strong evidence for increased miR-21 function in both models, as evidenced by the repressed targetomes (Figs. 1b-e).

- 2. There are several populations of macrophages present within the lung microenvironment. While Figure 1E shows the presence of these cells by staining with the macrophage marker CD68, it's not clear if there are differences in the total number, type of macrophage or activation status of these cells in their different models. In the supplemental figures they distinguish interstitial macrophages from alveolar macrophages via transcriptomics, but the field has moved beyond this to include ontogeny (monocyte derived vs. tissue resident) as a major contributor of macrophage function and activation. A detailed explanation of the role of miR-21 during different macrophage subpopulations as well as during injury/inflammation settings are needed.**

In both the COVID-19 and the bleomycin lungs, the total number of macrophages increased dramatically by approximately 3–4 fold (original Fig.1e). In response to the reviewer's question, we have now conducted additional analyses on human and mouse single cell datasets. These data demonstrate that dynamic changes took place in this cell type in both models. We then increased the granularity of the clusters of cells and dissected the macrophage population even further in both single cell datasets. The analysis revealed three distinct populations of macrophages: based on established known genetic markers and consistent with the recent literature¹⁸, we identified resident alveolar macrophages ($\text{Marco}^+ / \text{Mrc1}^{\text{hi}} / \text{Ccr2}^- / \text{Itgam}^{\text{lo}} / \text{Itgax}^{\text{hi}}$), recruited alveolar macrophages ($\text{Marco}^- / \text{Mrc1}^{\text{lo}} / \text{Ccr2}^{\text{hi}} / \text{Itgam}^{\text{hi}} / \text{Itgax}^-$) and interstitial macrophages ($\text{Marco}^- / \text{Mrc1}^{\text{lo}} / \text{Ccr2}^{\text{hi}} / \text{Itgam}^{\text{hi}} / \text{Itgax}^-$) as the main three different subclusters of macrophages. It can be seen that resident alveolar macrophages decreased in both models, whereas recruited alveolar macrophages and interstitial macrophages were significantly higher in both models. These findings suggest that recruited macrophages are crucial for responding to both COVID-19 and bleomycin inflammation, thus underscoring the importance of understanding their role in the immune response to these diseases. These data are **now presented as Supplementary Figure S3 and discussed in the revised manuscript text (page 4 lines 22-24).**

With regard to the role of miR-21 in different macrophage subpopulations, we kindly refer the reviewer to our answer to **reviewer 1, point 3 above**. In brief, our new data show increased miR-21 activity in all major pulmonary macrophage populations upon injury/inflammation.

- 3. In general, there was a lack of inflammation measured in this model. Although the investigators measured differences in other immune cells via scRNAseq and transcriptomics, there was no mention how this oligo treatment altered the number or activation of myeloid cells or other immune cells (such as neutrophils).**

We regret that we had not made this clear enough in the first version of our manuscript. Indeed, there is massive inflammation of the mouse lung in response to lung damage induced by bleomycin. This is evident by our data and is well documented in the recent literature. Several reports indicated a huge fraction of leukocyte populations in their single cell datasets obtained from lungs treated with bleomycin^{19,12}. Our data are in full agreement with these prior reports. We had carried out immunofluorescent staining for macrophages, the major population in the bleomycin-treated lungs, using an antibody against CD68, where we see the total number of macrophages increased dramatically by approximately 3–4 fold (original Fig.1e). As for our single cell study, we utilised a strategy similar to other groups (for example, Skelly et al. 2018²⁰) in that we de-enriched leukocytes prior to library

preparation and increased the footprint of non-leukocytes. In our study, we separated the leukocyte and non-leukocyte populations by magnetic sorting using an anti-CD45 antibody and mixed the two cell populations in a ratio of 1:1. This practice normalises the increase in leukocyte populations that is generally observed after bleomycin treatment (original Fig. S11b). However, the increase in leukocyte population can be seen in the original Fig. S11c, where we show a significant increase in the CD45+ fraction (approx. twice the number of cells) after bleomycin treatment, and this is normalised to that of PBS control levels in RCS-21-treated lungs. In a new analysis of the single cell dataset, we increased the granularity of macrophages into three different subclusters - resident alveolar, recruited alveolar and interstitial macrophages. We now show that treatment with RCS-21 decreased the fraction of recruited macrophages in response to bleomycin. **We have now included this as a new supplementary figure S12a-c.**

With regard to the activation of myeloid cells, we kindly refer the reviewer to original Figure S13a, where we have sequenced isolated primary alveolar macrophages from bleomycin-treated lungs compared to control treated lungs (upper left panel) and after RCS-21 treatment (upper right panel). The data clearly demonstrate a massive pro-inflammatory pattern of the alveolar macrophages after bleomycin-injury, whereas treatment with RCS-21 reversed the exaggerated pro-inflammatory response. In the revised manuscript, we have now – in addition to the new data as delineated above - made more clear the massive inflammatory response in the bleomycin model. First, we make the reader better aware that the normalisation procedure preceding the single cell RNA Seq experiments obscures the strong increase in inflammatory cells in the UMAP clustering graphs (**see revised methods section, Page 20, lines 24-25**). Second, we have **reworded the results sections on page 8 lines 12-17**.

Additional information regarding a target of how miR-21 is causing these effects in macrophages is needed.

We thank the reviewer for raising this important point. We have carried out new analyses that now demonstrate the effects of RCS-21 on the target mRNAs of miR-21. Our detailed analysis of the transcriptomic changes upon treatment with RCS-21 revealed that a relatively high fraction, around 70% of the entire miR-21 targetome (approx. 341 target mRNA with expression > 1 and containing a 7mer or 8mer binding site) was de-repressed upon treatment with RCS-21. While this is evident also from the right shift of the cumulative distribution curve in RCS-21 treated bleomycin-mice and SARS-CoV-2 infected slices (**original Fig. 5f and revised Supplementary Fig. S12e**), **we have now added a new Figure 5g**, where the respective expression changes are detailed for each individual mRNA target. These data clearly demonstrate, that rather than a single mRNA, it is the concerted effect of many miR-21 targets that constitutes the response to injury as well as to the treatment with RCS-21. GO term analysis was then employed to delineate the overall functional response mediated by the regulated mRNAs. Here, our data clearly demonstrate a strong pro-inflammatory signature that was significantly reversed upon treatment with RCS-21 in bleomycin lungs and SARS-CoV-2 infected human lung slices (**original Figs. S13a-b and 5i**).

We present the new data now in **Figs. 5g and revised Supplementary Fig. S12e and revised the text accordingly (page 8 lines 20-25, page 9 line 26)**.

- 4. The authors demonstrate that there is reduced lung elastance and compliance, decreased collagen accumulation and reduced expression of ECM related genes in the fibroblast cluster with treatment of RCS-21; however, there is no mechanism offered to explain how targeting macrophages could regulate fibrosis or collagen accumulation or expression.**

This is an interesting point that warrants more attention in this study. Indeed, we believe that the inflammatory macrophage is a key upstream factor for the pro-fibrotic activation of fibroblasts and this has been a long-term focus of our studies. We previously showed for cardiac fibrosis, that macrophage-specific genetic deletion of the miR-21 gene prevented fibrosis. Mechanistically, we had found that profibrotic, macrophage secreted factors mediated this effect²¹, including Spp1, Fn1 and Thbs1. Furthermore, recent data show that infection with SARS-CoV-2 triggers accumulation of monocyte-derived macrophages with a pro-fibrotic pattern²². In response to this reviewer's question, we conducted an intercellular communication analysis between macrophages and fibroblasts in response to bleomycin and the effect of RCS-21 on intercellular communication. After acute lung injury induced by bleomycin, strong changes in interaction (ligand receptor analysis determined by CellChat algorithm) were observed between interstitial macrophages and fibroblasts. Interestingly, we found the known pro-fibrotic mediators Spp1, Fn1 and Thbs1 to rank among the top regulated intercellular interactors, strongly suggesting them as bona fide mediators of macrophage-driven fibroblast activation. These paracrine interactions were repressed after RCS-21 treatment (**see reviewer figure 1**). As the precise delineation of the functional relevance of these interactions would require extensive new animal experimentation, we would prefer to not include these data in the present manuscript.

- 5. The authors make the point that miR-21 is upregulated specifically in macrophages, and that by generating a macrophage-targeted antimir-21 (RCS-21) this alters the development of lung fibrosis; however, in their PCLS culture they do not measure localization of miR-21 or presence of macrophages in their cultures. Yet, they observe significant differences in transcriptomics after infection with SARS-CoV-2. Is this due to RCS-21 affecting macrophages or other cellular populations? The relative role and possible downstream signalling pathways need to further explored.**

We agree with the reviewer on the importance of these points and have addressed them with new experiments. We stained hPCLS with an antibody against CD68, a macrophage marker, and could detect a high number of macrophages in these slices. To determine which cell type is taken up RCS-21 in these ex vivo cultures, we treated hPCLS with FAM-labelled RCS-21 and quantified the fluorescence signal in different cell types. RCS-21 was preferentially taken up by MRC1+ cells as opposed to other pulmonary cells types. GO term analysis was then employed to delineate the overall functional response mediated by the regulated mRNAs. Our data clearly demonstrate a strong proinflammatory signature that was significantly reversed upon treatment with RCS-21 in SARS-CoV-2 infected human lung slices (original Fig. 5i). However, we kindly make the reviewer aware that going deeper into the mechanism of these pathways would require extensive new experiments and is beyond the scope of this manuscript. We believe that the new data considerably strengthen our study and are presenting these **new results in a new Supplementary Figures S15 and S16 and changed the results part (page 9, lines 8-12) and the methods section (page 18 lines 10-16) accordingly.**

Minor:

- 1. The histological sections provided within the manuscript only show a very small section, larger sections of the entire lung or lobe should be shown in addition.**

In response to this reviewer, we now provide larger sections of the lung in addition to the higher magnification ones. Those are now taken with 20 x magnification. With even lower magnifications, the specific staining of the thin extracellular matrix structures is hardly visible. In addition, we are now showing a representative low magnification histological picture from a total of 12 mice (3 individual mice for each of the four groups). These data are presented in **new Supplementary figure S10d**.

- 2. There was no description regarding the sex of the mice utilized in this study. Additional information should be provided especially since there were relatively small samples sizes (n=3) included in the majority of the studies.**

We thank the reviewer of making us aware of this point and we added this information to revised methods section on **page 13, line 23**. Specifically, the Cx21 cKO (Fig. 1 f-i) and the RCS-21 experiments (Fig. 4a-e) were conducted on female mice.

- 3. It is not clear why the author chose to keep the bleomycin challenged mice in an 'O₂-enriched' cage for 14 days. Were the saline challenged mice also kept in these conditions? Has this been shown to alter fibrosis or recovery of the mice?**

Thank you for making us aware of this rather imprecise description. As recommended by our local animal regulatory authorities, we housed the experimental mice (all four groups) in closed IVC cages that received a moderate supplementation of oxygen, i.e. 30% final concentration for a relatively short period of time (days 5-9) after administration of bleomycin. In general, oxygen concentrations exceeding 70% for prolonged periods of time are considered to increase the risk of oxygen toxicity in animals²³. Thus, our transient and very moderate levels of oxygen are not expected to exert any toxic effect. As all groups received the identical treatment, we likewise can exclude a bias caused by oxygen supplementation. We have now clarified this point in the revised **methods section (page 14, lines 1-2)**.

- 4. Line 230-232 is an overstatement and needs to be reworded.**

We agree that this statement can in a narrow sense be interpreted as referring to data obtained from COVID patients, while we have obtained those data from mice. We therefore reworded the respective statement as follows:

Our data **in mice** suggest that RCS-21 is also effective beyond the immediate re-polarization of hyperactivated pulmonary macrophages and that it prevents pulmonary dysfunction and fibrosis (**page 11, line 22**).

References:

1. Cummings, R. D. The mannose receptor ligands and the macrophage glycome. *Curr. Opin. Struct. Biol.* **75**, 102394 (2022).
2. Martinez-Pomares, L. The mannose receptor. *J. Leukoc. Biol.* **92**, 1177–1186 (2012).
3. Riera, R. *et al.* Single-molecule imaging of glycan–lectin interactions on cells with Glyco-PAINT. *Nat. Chem. Biol.* **17**, 1281–1288 (2021).
4. Juliano, R. L. Intracellular Trafficking and Endosomal Release of Oligonucleotides: What We Know and What We Don't. *Nucleic Acid Ther.* **28**, 166–177 (2018).
5. Brown, C. R. *et al.* Investigating the pharmacodynamic durability of GalNAc–siRNA conjugates. *Nucleic Acids Res.* **48**, 11827–11844 (2020).
6. Maraganore, J. Reflections on Alnylam. *Nat. Biotechnol.* **40**, 641–650 (2022).
7. Humphreys, S. C. *et al.* Considerations and recommendations for assessment of plasma protein binding and drug–drug interactions for siRNA therapeutics. *Nucleic Acids Res.* **50**, 6020–6037 (2022).
8. Smith, E. S. *et al.* Clinical Applications of Short Non-Coding RNA-Based Therapies in the Era of Precision Medicine. *Cancers (Basel)*. **14**, 1–24 (2022).
9. Tay, M. Z., Poh, C. M., Rénia, L., MacAry, P. A. & Ng, L. F. P. The trinity of COVID-19: immunity, inflammation and intervention. *Nat. Rev. Immunol.* **20**, 363–374 (2020).
10. Walters, D. M. & Kleeberger, S. R. Mouse models of bleomycin-induced pulmonary fibrosis. *Curr. Protoc. Pharmacol.* 1–17 (2008) doi:10.1002/0471141755.ph0546s40.
11. Hay, J., Shahzeidi, S. & Laurent, G. Mechanisms of bleomycin-induced lung damage. *Arch. Toxicol.* **65**, 81–94 (1991).
12. Strunz, M. *et al.* Alveolar regeneration through a Krt8+ transitional stem cell state that persists in human lung fibrosis. *Nat. Commun.* **11**, 3559 (2020).
13. Delorey, T. M. *et al.* COVID-19 tissue atlases reveal SARS-CoV-2 pathology and cellular targets. *Nature* **595**, 107–113 (2021).
14. Melms, J. C. *et al.* A molecular single-cell lung atlas of lethal COVID-19. *Nature* **595**, 114–119 (2021).
15. Jackson, M. R. *et al.* Low-Dose Lung Radiation Therapy for COVID-19 Lung Disease: A Preclinical Efficacy Study in a Bleomycin Model of Pneumonitis. *Int. J. Radiat. Oncol. Biol. Phys.* **112**, 197–211 (2022).
16. Ali, A. S., Alrashedi, M. G., Ahmed, O. A. A. & Ibrahim, I. M. Pulmonary Delivery of Hydroxychloroquine Nanostructured Lipid Carrier as a Potential Treatment of COVID-19. *Polymers (Basel)*. **14**, 2616 (2022).
17. Saifi, M. A., Bansod, S. & Godugu, C. COVID-19 and fibrosis: Mechanisms, clinical relevance, and future perspectives. *Drug Discov. Today* **27**, 103345 (2022).
18. Aegerter, H., Lambrecht, B. N. & Jakubzick, C. V. Biology of lung macrophages in health and disease. *Immunity* **55**, 1564 (2022).
19. Aran, D. *et al.* Reference-based analysis of lung single-cell sequencing reveals a transitional profibrotic macrophage. *Nat. Immunol.* **20**, 163–172 (2019).
20. Skelly, D. A. *et al.* Single-Cell Transcriptional Profiling Reveals Cellular Diversity and Intercommunication in the Mouse Heart. *Cell Rep.* **22**, 600–610 (2018).
21. Ramanujam, D. *et al.* MiR-21-dependent macrophage-to-fibroblast signaling determines the cardiac response to pressure overload. *Circulation* **143**, 1513–1525 (2021).
22. Wendisch, D. *et al.* SARS-CoV-2 infection triggers profibrotic macrophage responses and lung fibrosis. *Cell* **184**, 6243–6261.e27 (2021).
23. Hochberg, C. H., Semler, M. W. & Brower, R. G. Oxygen toxicity in critically ill adults. *Am. J. Respir. Crit. Care Med.* **204**, 632–641 (2021).

Editorial Note: Figure has been redacted to maintain the confidentiality of unpublished data.

REVIEWERS' COMMENTS

Reviewer #1 (Remarks to the Author):

I find this version of the manuscript significantly improved. The authors should be commended for their excellent work at revising it.

Reviewer #2 (Remarks to the Author):

The authors have provided a sufficient detailed rebuttal including clarifications and additional experimentation. I have no further questions at this time.